# Demographically explicit scans for barriers to gene flow using gIMble

**Dominik R. Laetsch**[1]☉, **Gertjan Bisschop**[1]☉, **Simon H. Martin**[1],
**Simon Aeschbacher**[2], **Derek Setter**[1], **Konrad Lohse**[1]*

**1** Institute of Ecology and Evolution, University of Edinburgh, Edinburgh, United Kingdom, **2** Department of Evolutionary Biology and Environmental Studies, University of Zurich, Zurich, Switzerland

☉ These authors contributed equally to this work.
* konrad.lohse@ed.ac.uk

**Data Availability Statement:** All raw and mapped sequence data analysed by this project are previously published and are available from the European Nucleotide Archive (see Table A in S1 Text). All gIMble analysis commands used are available in the Supporting information. A jupyter

## Abstract

Identifying regions of the genome that act as barriers to gene flow between recently diverged taxa has remained challenging given the many evolutionary forces that generate variation in genetic diversity and divergence along the genome, and the stochastic nature of this variation. Progress has been impeded by a conceptual and methodological divide between analyses that infer the demographic history of speciation and genome scans aimed at identifying locally maladaptive alleles i.e. genomic barriers to gene flow. Here we implement genomewide IM blockwise likelihood estimation (gIMble), a composite likelihood approach for the quantification of barriers, that bridges this divide. This analytic framework captures background selection and selection against barriers in a model of isolation with migration (IM) as heterogeneity in effective population size ($N_e$) and effective migration rate ($m_e$), respectively. Variation in both effective demographic parameters is estimated in sliding windows via pre-computed likelihood grids. gIMble includes modules for pre-processing/filtering of genomic data and performing parametric bootstraps using coalescent simulations. To demonstrate the new approach, we analyse data from a well-studied pair of sister species of tropical butterflies with a known history of post-divergence gene flow: *Heliconius melpomene* and *H. cydno*. Our analyses uncover both large-effect barrier loci (including well-known wing-pattern genes) and a genome-wide signal of a polygenic barrier architecture.

## Author summary

As a fundamental process generating biological diversity, speciation involves the evolution of reproductive isolation and thus the build-up of barriers to genetic exchange among organismal groups. While population genomic data are arguably the only source of information we have about most recent speciation events, the way such data are analysed remains depressingly superficial: population genomic studies of speciation are phrased either as scans for outliers of genetic differentiation, or are based on models of neutral evolution under the constraint of a single genome-wide demography. Here we introduce a new statistical framework called gIMble to estimate the effective rate of gene flow and

notebook containing code/analyses used to generate all figures is available at https://github.com/LohseLab/gIMble/tree/master/example_output.

**Funding:** This work was supported by an ERC starting grant (ModelGenomLand, 757648) to KL which also supported DRL, GB and DS. KL was also supported by a Natural Environmental Research Council (NERC) UK Independent Research fellowship (NE/L011522/1) SHM was supported by the Royal Society URF/R1/180682. SA was supported by Swiss NSF grant SNF-31003A_182778. The funders had no role in study design, data collection and analysis, decision to publish, or preparation of the manuscript.

**Competing interests:** The authors have declared that no competing interests exist.

the effective population sizes along the genome from population genomic data. By capturing genome-wide variation in these two effective demographic parameters, gIMble disentangles the genomic footprints of different modes of selection and provides a direct quantification of the species barrier. To illustrate this framework, we analyse a classic speciation genomic dataset from *Heliconius* butterflies. We show that barriers to gene flow in this system include both large effect loci—most, but not all, of which were known from functional work—as well as a genome-wide signature of weak-effect polygenic barriers.

## Introduction

How reproductive isolation between species builds up at the genomic level remains a major question in evolutionary biology [1]. Population genomic studies inferring explicit demographic models of speciation for an increasing number of taxa suggest that species divergence in the face of gene flow is common. This poses a conundrum: gene flow homogenizes populations and—in the presence of recombination—leads to the breakdown of co-adapted combinations of alleles, which directly opposes divergent selection [2]. However, divergent selection for locally beneficial alleles, if sufficiently strong, may overcome the influx of maladapted migrant alleles [3, 4] at specific loci in the genome. It has been suggested that such barrier loci of large effect may act as nuclei around which further barriers can accumulate [5, 6]. The clustered genetic differentiation that may result from this feedback between migration, local selection and linkage has been referred to using metaphors of "speciation islands" [7] or "genomic islands of divergence" [6, 8]. Despite the fact that we now have access to vast volumes of whole genome sequence (WGS) data, it remains an open question whether the clustered genomic barriers envisaged by these verbal models are important in generating and maintaining new species in nature and, if so, how exactly they arise. More generally, it remains unclear how often species barriers are due to a few loci of large effect and how often they have a polygenic architecture. Progress on these questions has been stymied by a lack of methods that can extract from WGS data the relevant information about the population processes involved in the build up of barriers. Current speciation-genomic inference approaches can be classed into three broad categories [9]: **summary statistics scans**, **model-based demographic inference**, and **analyses of the relationship between recombination and divergence**. These have proven to be challenging to connect.

### Summary statistics scans

The search for speciation islands has led to an industry of genomic outlier scans that initially interpreted regions of increased differentiation between species as "speciation islands" [7, 8, 10, 11]. Outlier scans are generally based on summary statistics, in particular differentiation as measured by $F_{ST}$ [1, 12]. However, despite their conceptual appeal, genome scans in general and $F_{ST}$ scans in particular have been criticised on several grounds. Since $F_{ST}$ is a relative measure of divergence, it is sensitive to any force that locally reduces diversity. Thus, even in the absence of gene flow, $F_{ST}$ may be increased simply as a result of past positive selection [13, 14] and/or background selection (BGS) [13, 15] on linked sites [16, 17]. In fact, outlier scans trace back to Lewontin and Krakauer [18] who proposed a test for divergent selection based on the variance in $F_{ST}$. While a number of other summary statistics have been developed to overcome the limitations of $F_{ST}$ [19, 20], the fundamental problem of any scan based on a one-dimensional summary statistic is that it cannot distinguish between the alternative population genetic processes we wish to learn about [14, 21, 22]. In other words, it is unclear how often one

expects outliers in a summary statistic to arise simply by chance due to genetic drift or under scenarios that have nothing to do with the establishment of barriers to gene flow. As a result, distinguishing interesting outliers from noise has remained a major challenge [22].

## Model-based demographic inference

An arguably more successful line of research has been to use genome data to reconstruct the demographic history of speciation. A range of inference methods exists that fit more or less complex models of demographic history to whole-genome data (see [23–26] for reviews). An important starting point for quantifying gene flow during speciation has been the Isolation with Migration (IM) model [27], which assumes a constant rate of gene flow from the time of divergence until the present. Estimates of gene flow under the IM and related models have been obtained for many recently diverged species and have led to a reappraisal of gene flow as a ubiquitous ingredient of the speciation process [28, 29]. Over the past decade efforts have been made to extend multilocus demographic inference to model variation in effective migration rate ($m_e$) along the genome. While these methods have made it possible to diagnose heterogeneity in introgression globally by assuming that locus specific $m_e$ are drawn either from discrete bins or a continuous distribution [30–33], it has remained challenging to identify individual barrier regions.

## Modelling the relationship between recombination and divergence

A third type of speciation genomic analysis, which has so far only been applied to a small number of taxa, exploits the genome-wide relationship between genetic divergence and recombination to extract information about the selection on and density of barrier loci [9]. In contrast to the positive correlation between genetic diversity and recombination expected under background selection [15] (and globally positive selection), divergent selection against locally maladapted alleles results in a negative correlation between genetic divergence and recombination [34]. This prediction is based on theory showing that selection against deleterious immigrant alleles reduces the effective rate of gene flow at linked neutral sites [2] below the rate $m$ at neutral sites that are not affected by barriers. The extent of this reduction can be captured by the effective migration rate $m_e$, which decreases with the strength of selection and increases with the map distance from selective targets [2, 35]. Integrating over the unknown genomic positions of barrier loci and conditioning on a recombination map, Aeschbacher et al. [9] developed an approach for jointly estimating the aggregate strength of selection against immigrant alleles in the genome, the divergence time, and the baseline migration rate. However, although this method uses the expected correlation between recombination and genetic divergence under a demographically explicit model, it also infers genome-wide parameters, and so it is uninformative about the genomic locations of barrier loci.

## Demographically explicit genome scans

Given the current state of inference approaches, speciation genomic analyses involve an awkward dichotomy: one either infers a demographic history, which—however detailed—is uninformative about which genomic regions may act as barriers. Alternatively, one may visualise the heterogeneity in differentiation along the genome via summary statistics, which partially sacrifices the ability to learn about the population processes that have given rise to this heterogeneity. While it is common practice to analyse speciation genomic data both in terms of demography and genome scans, these analyses currently remain separate. Thus, interpreting the results of genome scans in the light of an inferred demography is a qualitative, *post hoc* step rather than quantitative exercise.

Ideally, we would like to reconstruct a fully specified model of the speciation process, including both the demographic history—parameterised in terms of genetic drift, divergence, and gene flow—as well as all relevant selective forces (i.e. a plausible null model *sensu* [36]). Indeed, since most selection leaving a footprint in genomes is likely unrelated to the emergence of reproductive barriers [37], even a very simple cartoon of the speciation process would have to capture both selection on variation that is globally beneficial or deleterious, and local selection that establishes and maintains barriers to gene flow. Unfortunately, it is currently impossible to infer such a detailed model from genome data.

A simpler approach is to model the selective processes that lead to the establishment of barriers to gene flow via heterogeneity in effective migration rate $m_e$. This approach has been pioneered by Roux et al. [38] and has been implemented as an approximate Bayesian computation framework [39] that allows fitting models of heterogeneous gene flow to genome-wide data. However, while this method estimates variation in $m_e$ (and $N_e$) via aggregate genome-wide distributions, it does not estimate local $m_e$ along the genome.

The motivation of the present study is to fill this gap and develop a framework based on likelihood calculation that allows interpreting the heterogeneity of divergence and diversity along the genome through the lens of an explicit, albeit necessarily simple, demographic history. We approximate the effects of background selection (BGS) and selection against migrants as variation in effective population size ($N_e$) and effective migration rate ($m_e$), respectively. Our method infers genome-wide histories of isolation with migration (IM) using a block-wise likelihood calculation. The composite likelihood calculation at the heart of our approach is based on previous analytic work [40–42] and is implemented as part of a modular, open source bioinformatic toolset: genome-wide IM blockwise likelihood estimation `gIMble` v.1 (https://github.com/LohseLab/gIMble). `gIMble` allows for an efficient inference of $m_e$ and $N_e$ in windows along the genome as a scan for barriers to gene flow, includes algorithms for the required partitioning, filtering and summarizing of genomic variation, and provides a simulation module for generating parametric bootstraps using the coalescent simulator `msprime` [43].

### *Heliconius melpomene* and *H. cydno* as a test case

Throughout the paper, we describe our approach for demographically explicit scans for barriers to gene flow by way of a re-analysis of data from two sister species of *Heliconius* butterflies: *H. melpomene* and *H. cydno*. This classic model of speciation research provides an excellent test case for our approach given that the genome-wide heterogeneity in divergence and diversity has been studied extensively in this species pair [44–50] and the genetic basis of key barrier traits (wing patterns and preference for them) is known [45–47, 51]. Despite strong behavioural, ecological, and postzygotic isolation (female hybrid sterility), there is overwhelming evidence for ongoing hybridisation and gene flow between these species that affects a large proportion of the genome [41, 46]. Martin et al. [46] found that $F_{ST}$ between *H. melpomene* and *H. cydno* is highly heterogeneous along the genome, but concluded that this heterogeneity does not provide direct evidence of barriers to introgression. Indeed, $F_{ST}$ between *H. melpomene* and *H. cydno* is similarly heterogeneous and strongly correlated in both sympatric and allopatric comparisons, despite the fact that allopatric populations are thought to experience no contemporary gene flow [46]. Many $F_{ST}$ outliers between *H. melpomene* and *H. cydno* may therefore simply reflect increased rates of within-species coalescence [14]. Using this well studied *Heliconius* system as a test case, we showcase `gIMble` and address the following questions:

i) What fraction of the genome likely acts as a complete barrier ($m_e = 0$) between these taxa? ii) What is the overlap/correlation between barriers defined via $m_e$ and $F_{ST}$ outliers? iii) Do $m_e$ barriers include known true positives, i.e. major-effect loci controlling phenotypes involved in

reproductive isolation? iv) Is there a genome wide relationship between $m_e$ and the rate of recombination, as predicted by models that assume polygenic barriers?

## Results

### The model

The framework implemented in `gIMble` models the genomic landscape of speciation by constructing scans of two kinds of effective demographic parameters: background selection (BGS) is captured via heterogeneity in $N_e$, and barriers to gene flow are modeled via heterogeneity in $m_e$. Unlike previous approaches [9] which assume a relationship between $m_e$ and recombination, our inference does not rely on knowledge of the recombination map or any assumed relationship between recombination and $m_e$.

We focus on an IM history between two species A and B (Fig 1A). We assume divergence at a time $T$ in the past (measured in generations) and a constant probability of unidirectional migration (see Discussion for the effect of bi-directional migration), i.e. lineages sampled in species B may trace back to A via migration at rate $m_e$. Effective population sizes are allowed to differ between species, i.e. the most complex model assumes three different $N_e$ parameters: $N_A$, $N_B$ and $N_{anc}$.

One could in principle co-estimate all five parameters of the IM model ($N_A$, $N_B$, $N_{anc}$, $m_e$, $T$) in sliding windows along the genome. However, in practice, this maximally complex model in which all parameters vary freely along the genome is both hard to infer—given that IM parameters are known to have correlated errors [27]—and difficult to interpret. Thus, we conduct a two-step inference (Fig 1): first, we identify the best-fitting *global* IM history that assumes no heterogeneity in demographic parameters along the genome. In a second step, *local* variation in $N_e$ and $m_e$ is estimated in sliding windows along the genome. This inference is conditioned on the estimate of $T$ obtained under the global model, i.e. we assume that the onset of species divergence is a global event that is shared by the whole genome. Note that, unlike the approach of Fraïsse et al. [33], `gIMble` models variation along the genome in $m_e$ rather than the scaled migration rate $M = 4N_e m_e$, which may vary simply as a function of $N_e$.

### Summarising genomic variation in blocks and windows

The basic unit of data in the inference implemented in `gIMble` is a block of sequence of a fixed number of sites in which we record sequence variation in a heterospecific pair of individuals. We will refer to this as a "pair-block". Crucially, we assume that blocks are only indirectly affected by selection at nearby linked sites and otherwise evolve neutrally (the *Heliconius* analyses described below are restricted to intergenic data) and under a constant mutation rate (see Discussion for the effect of $\mu$ heterogeneity on inference). While the assumptions of neutrality and a fixed block length and mutation rate allow us to treat blocks as statistically exchangeable, they necessitate a careful and consistent filtering strategy for variant and invariant sites. In particular, only sites (both variant and invariant) that meet coverage criteria contribute to the length of a pair block. As a consequence, the physical span of blocks may be longer than the block length (Fig 1B). Similarly, windows, defined in terms of a fixed number of pair-blocks, typically span sequence that is excluded from the analysis, either because of its genomic annotation (e.g. coding sequence, repeats) or because coverage criteria are not met (Fig 1D). An important property of our blocking algorithm is that pair-blocks are defined independently for each pair of heterospecific individuals. This avoids sample-size dependent filters, e.g. we do not restrict inference to data that meet filtering criteria in a fixed fraction of individuals. To

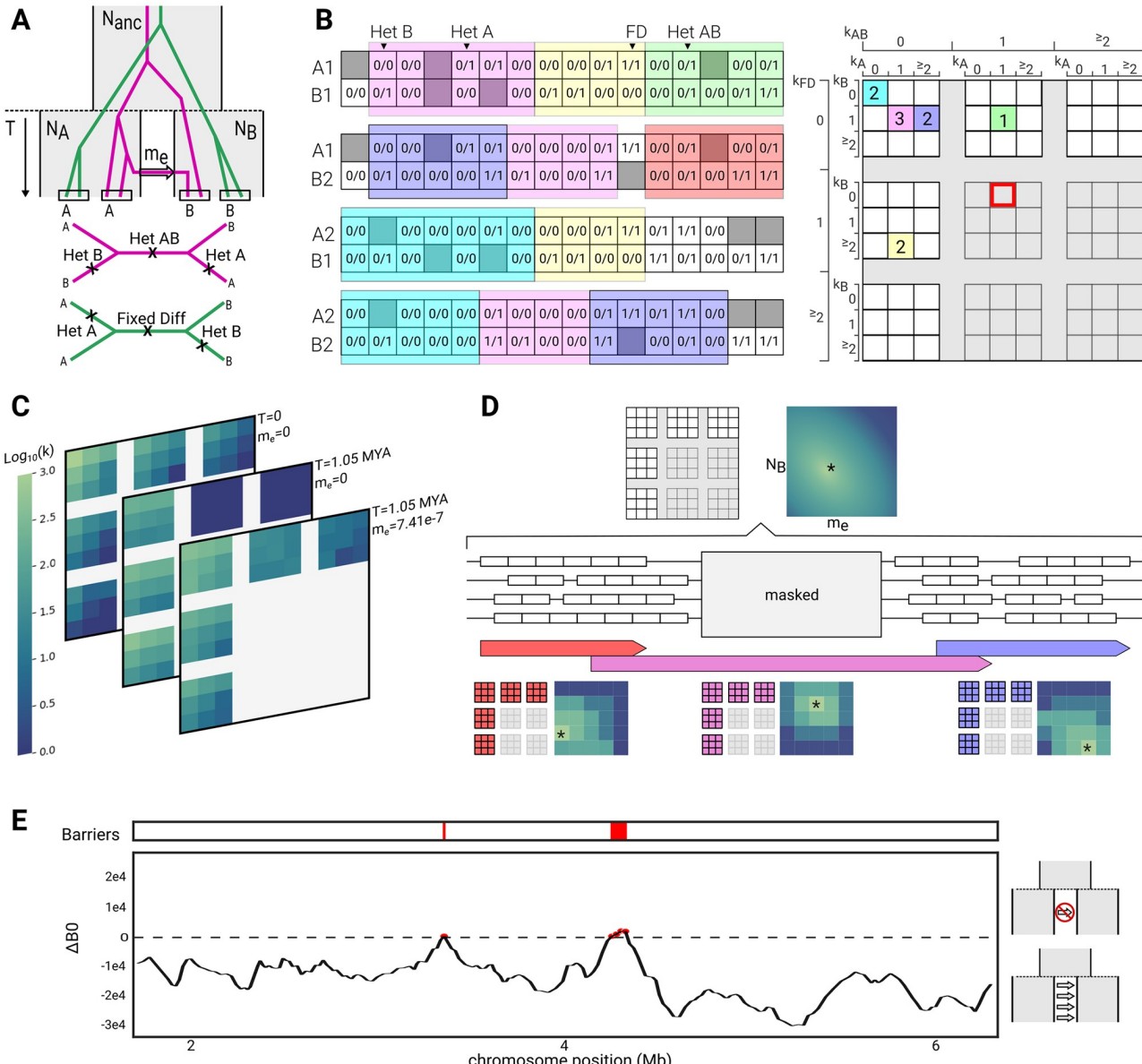

**Fig 1. Demographic inference using `gIMble`.** A) The 5-parameter IM model assumes two populations (A and B) with effective sizes $N_A$ and $N_B$ that split $T$ generations in the past from a common ancestral population with size $N_{anc}$ and experience unidirectional gene flow at rate $m_e$. Pairs of diploids are sampled, one from each population, resulting in two distinct possible topologies. Assuming an infinite sites mutation model, mutations that are heterozygous in both A and B (Het AB) are exclusive to the magenta topology, while fixed differences (Fixed Diff) are exclusive to the green. Mutations on A and B branches may occur on either topology, creating heterozygous variants in either the A (Het A) or the B (Het B) individual. B) Pair-blocks are created from left to right, each containing a fixed number ($l$) of sites callable in both individuals. The span of any pair-block may be longer than $l$ due to missing data (gray squares). Pair-blocks are summarised by the number of each type of mutation they carry (the bSFS). The number of occurrences of each type of pair-block is recorded in a four-dimensional array (the bSFS tally). Pair-blocks that violate the four-gamete test (e.g. shaded in red) are excluded from the analysis. C) The expected bSFS tally depends on the underlying model and parameter values. Divergence without gene flow skews the bSFS tally toward higher frequencies of the green topology (and thereby fixed differences) and generally increases divergence. Higher levels of migration or a shorter split time increase the frequency of the magenta topology (and thereby shared heterozygous sites). D) Pair-blocks are made along the genome for all possible pairs of individuals. Small gaps between pair-blocks represent missing data while large gaps are regions masked by the user (e.g. genes or repeats). The bSFS tally from the entire genome is used to identify the most likely parameters of the (global) IM model. In a subsequent step, sliding windows of a fixed number of pair-blocks (here, 20) and a fixed overlap (here, 5) are made (the red, purple, and blue arrow-headed bars). The span of a window depends on the physical location of the pair-blocks from which it is composed. For each window, a grid of parameter values is searched to infer the combination of local parameters that maximises the corresponding composite likelihood. E) the support for barriers to gene flow for each window can be quantified by comparing the fit to the global model relative to a model of locally reduced gene flow.

facilitate comparability across datasets, we have implemented a minimal set of standard filters within the `gimbleprep` module.

Inference is based on the block-wise site frequency spectrum (the bSFS *sensu* [41]), i.e. the tallies of SFS mutation types in blocks. The SFS of a pair-block is simply a vector of counts of the four possible types of mutations: heterozygous sites in A (Het A), heterozygous sites in B (Het B), shared heterozygous sites (Het AB) and fixed differences (Fixed Diff) (Fig 1A). The bSFS tally, which can be defined for the entire dataset or for a window consisting of a large number of pair-blocks, is a (four-dimensional) tally of SFS vectors (Fig 1B).

Lohse et al. [40] showed how to analytically compute the probability of SFS configurations under the IM and related demographic models in the absence of recombination using the generating function (GF) of genealogies. `gIMble` implements this calculation using `agemo` [42] and supports both global and local inference through a number of modules. The `optimize` module uses the bSFS data to find maximum composite likelihood estimates (MCLE) under the IM model using standard numeric optimization (Fig A in S1 Text, (3)). In contrast, `gridsearch` infers IM parameters in sliding windows over a grid of bSFS probabilities pre-computed using `makegrid` (Fig A in S1 Text, (4) & (5)). Importantly, to quantify the uncertainty in parameter estimates, `gIMble` includes a module to `simulate` parametric bootstrap replicates under the inferred demographic history. See Fig A in S1 Text and Methods for a detailed overview of the `gIMble` modules.

## A butterfly test case

To illustrate `gIMble`, we re-analysed WGS data of 10 wild-caught individuals each of *H. melpomene* and *H. cydno*, sampled from sympatric populations in Panama (of the subspecies *H. melpomene rosina* and *H. cydno chioneous*, respectively (Table A in S1 Text)). To maximise the density of variable sites and to satisfy the assumption of selective neutrality, we focused all analyses on intergenic sequence, which we defined as any sequence without gene or repeat annotation. For the sake of simplicity, we excluded the Z chromosome, which is known to have greatly reduced genetic diversity and gene flow in *Heliconius* taxa [46, 50]. Thus, while it would be interesting to include the Z in a `gIMble` analysis of species barriers, in practice, this would exclude female samples. More importantly, in the absence of information about the relative mutation rates in males and females and the effective sex ratio, interpreting Z-to-autosome differences is notoriously difficult.

After filtering for coverage and quality, we partitioned the data into pair-blocks of length $l = 64$ bases. Given that the likelihood calculation assumes no recombination within blocks, the choice of block length involves a trade-off between power (which increases with $l$) and bias (which also increases with $l$). We explored the effect of block length on parameter estimates (see next section) and chose $l = 64$ for most analyses. This block length corresponds to an average of one heterozygous site (in each taxon) per block ($l^*\pi \sim 1$), which is a practical compromise for organisms that have similar rates of mutation and recombination $r_{bp}/\mu \sim 1$. This ensures that around half of all pair-blocks contain multiple variants (Fig B in S1 Text). The total block-wise dataset comprises a mean length of 72 Mb of sequence per heterospecific sample pair. This corresponds to 69% of the total intergenic sequence and 24% of the entire *H. melpomene* genome. We find that the average heterozygosity ($H$) in intergenic blocks within each species (Table 1) is comparable to the mean pairwise genetic diversity [52] across all sample pairs ($\pi = 0.0163$ and $0.0156$ in *H. cydno* and *H. melpomene*, respectively), which suggests little evidence for inbreeding or local population structure.

We summarised blockwise genetic variation in windows of 50,000 pair-blocks with a 20% offset between adjacent windows, that is, overlapping windows are shifted by 10,000 pair-

**Table 1. Average heterozygosity (*H*) across all individuals within each species and divergence (*d*$_{xy}$) between *H. melpomene* and *H. cydno*.** Summaries were generated using `gIMble info` based on intergenic blocks of 64 bases sampled in heterospecific pairs of individuals (top row). Estimates from a similar previous analysis for a single pair of individual genomes and longer blocks of 150 bases [41] (second row) are shown for comparison.

| Study | Data | $H_{Hcyd}$ | $H_{Hmel}$ | $d_{xy}$ | $F_{ST}$ |
|---|---|---|---|---|---|
| This study | 64 bases | 0.0158 | 0.0155 | 0.0220 | 0.169 |
| Ref. [41] | 150 bases | 0.0169 | 0.0150 | 0.0218 | 0.155 |

blocks. Given that we sampled 10 individuals from each taxon (i.e. there are 100 possible hetero-specific sample pairs), this window size corresponds to a minimum window span of $500 \times 64$ bases = 32 kb of "blocked" sequence if we assume complete coverage across all 100 hetero-specific sample pairs. However, since we only considered intergenic sequence that met coverage and quality filters, the span (distance between first and last base) of windows is considerably greater (median 103 kb, Fig C in S1 Text). Exploring window-wise variation, we recover several known properties of this dataset: first, outliers of $F_{ST}$ are mostly due to reduced diversity $\pi$ rather than increased absolute divergence $d_{xy}$ (Fig D in S1 Text). Second, differentiation along the genome is highly variable and includes many peaks, but no large genomic regions of elevated $F_{ST}$ (Fig E in S1 Text). Finally, $F_{ST}$ is negatively correlated with recombination (Fig F in S1 Text).

## Fitting a global background model of divergence and gene flow

Our initial aim was to identify a plausible, yet simple, global background demography for this *Heliconius* species pair. We compared support for IM models with gene flow in either direction, as well as simpler nested histories. Specifically, we fitted a history of strict divergence (*DIV*, $m_e = 0$) and a model of migration only (*MIG*, $T = \infty$) (Table B in S1 Text).

The scenario that best fit the global bSFS is an IM model involving gene flow at rate $m_e = 7.41 \times 10^{-7}$ (which corresponds to $M = 4N_{mel}m_e = 1.62$ individuals) per generation from *H. cydno* into *H. melpomene* (forwards in time) (*IM*$_{\rightarrow mel}$ Table 2, Fig G in S1 Text). We refer to this model as "the global model" from here on. Our estimates under the global model are consistent with both previous demographic analyses for this species pair [41, 45] (Table 2). Assuming four generations per year and a spontaneous mutation rate estimated for *Heliconius* of $2.9 \times 10^{-9}$ per bp and generation [53], the maximum composite likelihood estimates (MCLE) under the best fitting IM model correspond to effective population sizes ($N_e$) of roughly $0.5 \times 10^6$, $1.5 \times 10^6$, and $1 \times 10^6$ for *H. melpomene*, *H. cydno* and their shared ancestral population, respectively, and a divergence time of approximately 1 MYA (Fig G in S1 Text, Table 2). These parameter estimates are robust to the choice of block length, i.e. partitioning the data into shorter (48 bases) or longer (128) blocks gives comparable parameter estimates (Table C in S1 Text). Note that the second best model, *MIG* ($T = \infty$) also gives very similar estimates of $m_e N_e$ (see Table B in S1 Text and Discussion for the robustness of the *Heliconius* analysis to model choice).

**Table 2. Divergence and gene flow estimates for *H. melpomene* and *H. cydno* inferred using `gIMble optimise`.** By default `gIMble` parameter estimates are scaled in absolute units, i.e. number of individuals and generations given a user-specified $\mu$. We have converted *T* to millions of years (MYA) assuming four generations per year. 95% CI obtained from a parametric bootstrap were: $[5.36 \times 10^5, 5.63 \times 10^5]$, for $N_{mel}$, $[1.40 \times 10^6, 1.43 \times 10^6]$ for $N_{cyd}$, $[9.25 \times 10^5, 9.31 \times 10^5]$ for $N_{anc}$, $[1.046, 1.061]$ for *T* and $[7.30 \times 10^{-7}, 7.53 \times 10^{-7}]$ for *m*. The bottom row gives estimates from a previous analysis [41] that was based on a single pair of individual genomes for comparison.

| model/dataset | $N_{mel}$ | $N_{cyd}$ | $N_{anc}$ | $T$ (MYA) | $m$ ($M = 4N_e m$) |
|---|---|---|---|---|---|
| $IM_{\rightarrow mel}$ | $5.49 \times 10^5$ | $1.415 \times 10^6$ | $9.279 \times 10^5$ | 1.054 | $7.41 \times 10^{-7}$ (1.62) |
| $IM_{\rightarrow mel}$ [41] | $1.10 \times 10^6$ | $2.85 \times 10^6$ | $N_{cyd}$ | 1.04 | $3.41 \times 10^{-7}$ (1.50) |

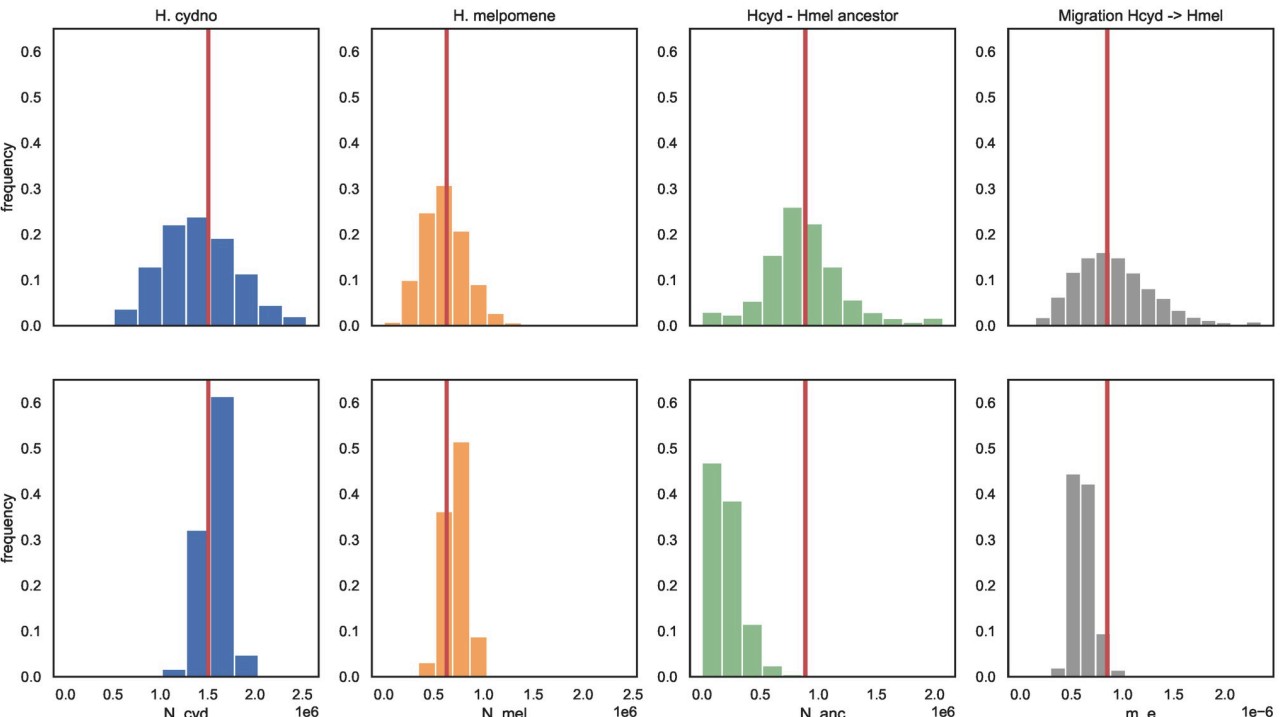

**Fig 2. Effective demographic parameters in *H. cydno* and *H. melpomene*.** The distribution of $N_{mel}$, $N_{cyd}$, $N_{anc}$ and $m_e$ from *H. cydno* into *H. melpomene* estimated in sliding windows (of span $\sim$ 100kb) for the real data (top row) and bootstrap simulations without $m_e$ and $N_e$ heterogeneity (bottom row). Simulations were conditioned on the *Heliconius* recombination map [54] (see Methods). The red lines indicate estimates under the global model; histogram bins correspond to points in the $12 \times 12 \times 12 \times 16$ parameter grid used for inference.

## Strong support for heterogeneity in $N_e$ and $m_e$

We explored variation in $N_{mel}$, $N_{cyd}$, $N_{anc}$ and $m_e$ along the *Heliconius* genome by exhaustively searching parameter combinations in a pre-computed $12 \times 12 \times 12 \times 16$ grid. The grid was roughly centred on the MCLEs inferred under the global model and $T$, the species divergence time, was fixed to the global estimate (Table 2). The choice of grid involves a trade-off between resolution and computational resources and—in practice—is most likely an iterative procedure that depends on the variation in parameter estimates in the data. However, instead of opting for a grid that is as fine as our computational resources would allow, we chose bounds that capture most of the variation in parameter estimates and a resolution in the $m_e$ direction that accommodates our barrier definition.

We find that window-wise, i.e. local, MCLEs of all three $N_e$ parameters vary by a factor of three (Fig 2). Similarly, local estimates of $m_e$ have a wide distribution. Importantly, applying the same inference to window-wise data simulated under an IM model without heterogeneity in effective demographic parameters (i.e. the MCLEs under the global model, Table 2) gives much narrower distributions of local estimates (Fig 2, bottom). Thus, the heterogeneity in local $N_e$ and $m_e$ can neither be explained by variation in recombination rate alone, nor by the randomness of the coalescent and the mutational process (all of which were included in the simulation), but likely captures true variation in $N_e$ and $m_e$ due the varying effects of selection along the genome.

We can quantify the overall support for heterogeneity in $N_e$ and $m_e$ by summing the log composite likelihood (ln *CL*) across windows. In particular, the support for a maximally

**Table 3. Model fit measured as $\Delta \ln CL$ relative to the best fitting global model, $IM_{\rightarrow mel, fix}$ (Table 2) which assumes that all effective parameters are fixed.**

| Model | $DIV_{fix}$ | $MIG_{\rightarrow mel, fix}$ | $IM_{\rightarrow mel, fix}$ | $IM_{\rightarrow mel,hetN_e}$ | $IM_{\rightarrow mel,hetN_e,m_e}$ |
|---|---|---|---|---|---|
| $\Delta \ln CL$ | -1,057,371 | -194,281 | 0 | 3,933,396 | 5,103,312 |

heterogeneous model that allows for variation in $N_{mel}$, $N_{cyd}$, $N_{anc}$ and $m_e$ is the sum of $\ln CL$ associated with all local MCLEs. Allowing for heterogeneity in all three effective parameters greatly improved the overall fit of the $IM_{\rightarrow mel}$ model (Table 3).

## Detecting barriers to gene flow

Estimating $m_{e,i}$ for each window $i$ (shorthand $m_i$) along the genome immediately allows us to detect genomic barriers as regions with reduced effective migration. One possibility is to interrogate the point estimates of $m_i$ along the genome. However, in quantifying the support for reduced $m_e$, we want to account for the confounding effects of locally varying effective population sizes $N_i$. To this end, we define $\Delta_B$ as the relative support for a model in which $m_i$ is reduced below some threshold $m_e^*$ relative to a model in which $m_i$ is fixed to $\hat{m}_e$, the value that best fits the data under the global model assuming no heterogeneity in $m_e$. That is,

$$\Delta_B = \max_{m_i \leq m_e^* \in \mathbf{M}} \left\{ \ln CL(m_i, \hat{T}, N_i) \right\} - \max_{N_i \in \mathbf{N}} \left\{ \ln CL(\hat{m}_e, \hat{T}, N_i) \right\}.$$

$$N_i \in \mathbf{N} \tag{1}$$

Here, $\hat{m}_e$ and $\hat{T}$ are the best supported parameter estimates globally and $\mathbf{M}$ and $\mathbf{N}$ are the grids of $m_e$ and $N_e$ values used for the window-wise optimization. The first term represents the highest support ($\ln CL$) obtained for window $i$ across all possible population sizes $N_i$ and migration rates $m_i$ below the threshold value $m_e^*$. The second term gives the support of the best fitting model when the migration rate is fixed to the global estimate $m_i = \hat{m}_e$ and only $N_i$ may vary. We can think of $\Delta_{B,0} = 0$ as defining a sea level for barriers to gene flow, i.e. windows with a positive value have support for reduced gene flow, windows associated with a negative value of $\Delta_{B,0}$ span genomic regions for which a history that includes gene flow at background level $\hat{m}_e$ fits better than a complete barrier model.

Given that `gIMble` analyses are restricted to putatively neutral sequence, i.e. barrier loci are unlikely to contribute directly to the bSFS tally of a window, we expect $m_i > 0$ even for a window that spans a strong/complete barrier locus. We therefore explored the *Heliconius* dataset using two $m_e$ thresholds to define barriers, $m_e^* = 0$ or $m_e^* = 0.2 * \hat{m}_e$ (we refer to these analyses as $\Delta_{B,0}$ and $\Delta_{B,0.2}$ scans respectively). Only 0.78% of windows (87 out of 11,217) meet the strictest barrier definition $\Delta_{B,0} > 0$. We merged all overlapping barrier windows above the "sea-level" of $\Delta_{B,0} = 0$ which defines 25 barrier regions of a total length of 4.96 Mb (1.9% of autosomal sequence). Individual barrier regions range in length from 90 to 408 kb (Fig 3, top). A less stringent barrier definition, $\Delta_{B,0.2} > 0$, identifies 7.8% of windows as barriers, spanning 170 barrier regions of a total length of 47.4 Mb (18.2% of autosomal sequence) (Fig 3, bottom).

Since the composite-likelihood measure of barrier support $\Delta_B$ depends both on the locally best fitting set of $\mathbf{N}$ and the local recombination rate, we used a parametric bootstrap (based on window-wise simulation replicates generated with `gIMble simulate`, see Methods) to estimate window-wise false positive rates (FPR). We find that the average genome-wide FPR are far lower than the observed fraction of barrier windows: 0.017% and 0.44% for $\Delta_{B,0}$ and $\Delta_{B,0.2}$ scans, respectively (Fig 3).

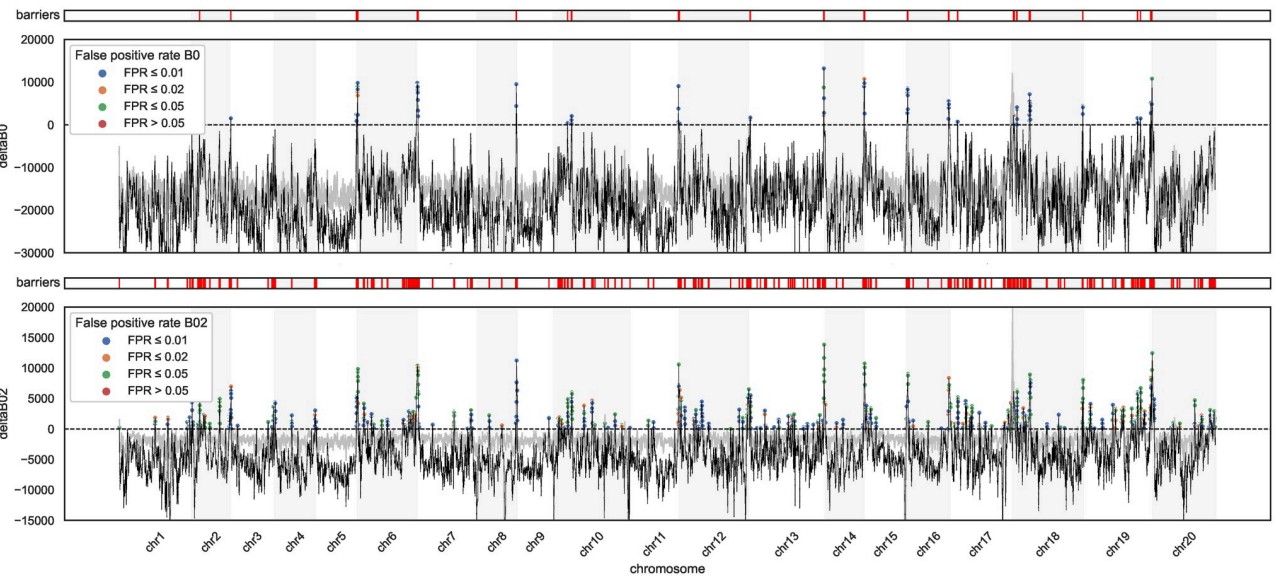

**Fig 3. Barriers to gene flow between _H. melpomene_ and _H. cydno_ inferred using `gIMble`.** The $m_e$ threshold used to diagnose barriers is relaxed from top ($\Delta_{B,0}$) to bottom ($\Delta_{B,0.2}$). Regions above the "sea-level" of $\Delta_B = 0$ fit a history of reduced $m_e$ better than a model assuming the global estimate $\hat{m}_e$. Windows with $\Delta_B > 0$ have been coloured to reflect their expected FPR. The $\Delta_B$ threshold corresponding to a FPR of 0.05 is shown in grey.

## Barriers only partially overlap $F_{ST}$ outliers but coincide with known wing-pattern genes

Although $F_{ST}$ and $\Delta_B$ are correlated (e.g. for $\Delta_{B,0}$, Pearson's $\rho = 0.783$, $p < 0.000001$, Fig 4), we find that $\Delta_B$ barriers only partially overlap $F_{ST}$ outliers. This is true irrespective of how $\Delta_B$

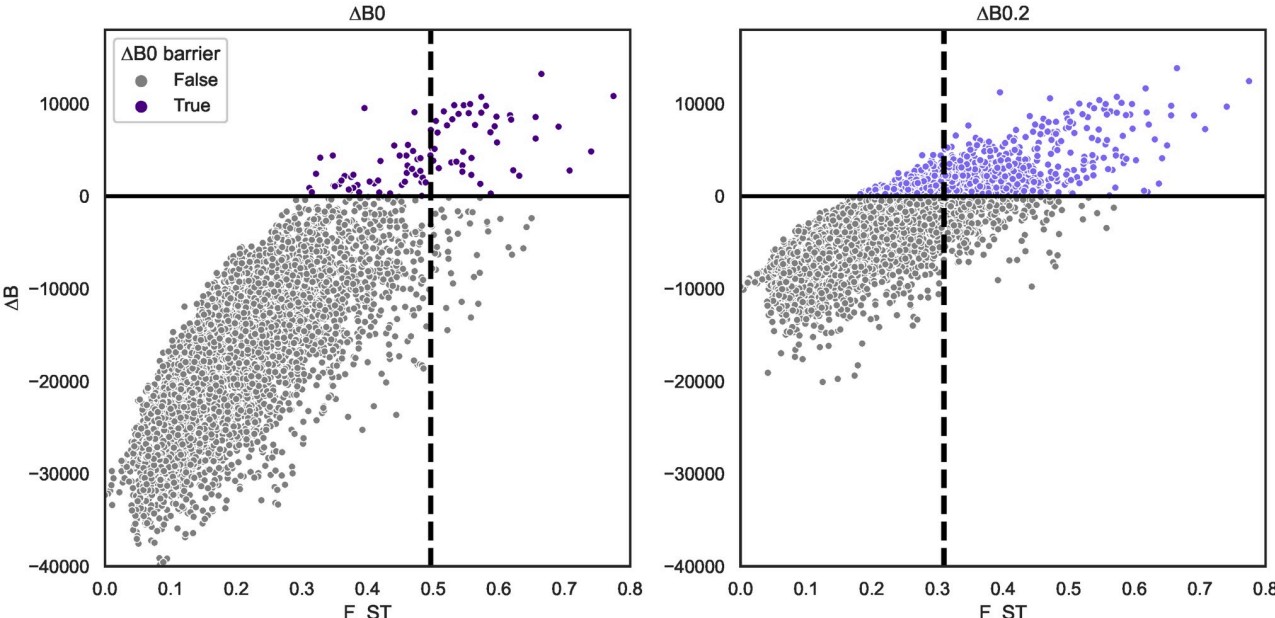

**Fig 4. $F_{ST}$ and support for barriers to gene flow between _H. melpomene_ and _H. cydno_ as measured by $\Delta_B$ (Eq 1) are partially correlated.** Each dot represents an autosomal window of span ~100 kb (Fig C in S1 Text). Two $m_e$ thresholds used to distinguish barriers are shown: $\Delta_{B,0}$ (left) and $\Delta_{B,0.2}$ (right) which identify 0.78% and 7.8% of windows as barriers respectively. The dashed vertical lines delineate the corresponding percentiles in the $F_{ST}$ distribution.

barriers are defined. For example, only 42 of the 87 $\Delta_{B,0}$ barrier windows are contained in the analogous $F_{ST}$ tail (Fig 4). In other words, our demographically explicit scan identifies novel and different genomic regions which are not necessarily outliers in $F_{ST}$.

To investigate whether the $\Delta_B$ scan recovers previously identified barrier loci, we focused on the three large-effect genes controlling wing-pattern differences between *H. melpomene* and *H. cydno* that have been studied in detail [55–57]: *wnt-A* (chromosome 10), *cortex* (chromosome 15), and *optix* (chromosome 18). We find that all three of these known positives are barriers as defined by $\Delta_{B,0.2}$ and two (*wnt-A* and *optix*) are also barriers under the most stringent criterion of $\Delta_{B,0} > 0$ (Fig 5). While *optix* is also contained in the analogous tail of $F_{ST}$ outliers (i.e. the 0.78% of windows with highest $F_{ST}$), both *wnt-A* and *cortex* are not. Intriguingly, the two $\Delta_{B,0}$ barrier regions downstream of *optix* on chromosome 18 coincide with a recently identified QTL peak of species-specific visual preference behaviours for the red wing-pattern controlled by *optix* [44]. This region includes five candidate genes [58], specifically cis regulatory differences in *Grik2* and *regucalcin2* (highlighted in orange in Fig 5).

## Barriers are concentrated in regions of low recombination and high coding density

Population genetic models of species barriers predict a reduction of $m_e$ in the vicinity of barrier loci. This reduction scales with the the map distance from the barrier locus and the strength of divergent selection acting on it [9, 61, 62]. In regions of low recombination there is thus an increase both in the physical scale at which barrier loci reduce gene flow in flanking neutral regions and in the chance that new barrier loci arise [63, 64]. However, since the effect of any form of selection on nearby neutral sequence depends on recombination, quantifying the association between barriers and $r_{bp}$ has proven hard using summary statistics (but see [65]). Thus, $F_{ST}$ outliers are expected to be concentrated in regions of low recombination simply as a result of the ubiquitous positive correlation between genetic diversity and recombination rate caused by BGS and positive selection [13, 66].

In contrast, our framework of estimating effective parameters and defining barriers of $m_e$ via $\Delta_B$ allows us to disentangle selection acting on barriers from BGS and positive selection not contributing to species differentiation. Likewise, since our inference of $m_e$ and barriers does not make use of information on recombination, we can explore the relationship between $r_{bp}$ and $m_e$ over different physical scales in a second step. After controlling for heterogeneity in $N_e$, we find a clear association between local $m_e$ and direct estimates of recombination rate ($r_{bp}$) over two physical scales. At an inter-chromosomal scale, $m_e$ is negatively correlated with physical chromosome length (Fig 6, Pearson's $\rho = -0.63$, $p = 0.0073$). This pattern is expected under polygenic selection against interspecific gene flow because longer chromosomes tend to have lower $r_{bp}$ in *Heliconius* [67]. A negative correlation between chromosome length and introgression—measured in terms of $f_d$, a summary statistic that draws information from taxon trios—has also been found in a previous study of the same *Heliconius* species pair [65]. Reassuringly, we find estimates of local $N_e$ (both $N_{cyd}$ and $N_{mel}$) to be even more strongly correlated with chromosome length (Fig H in S1 Text) than $m_e$, as is expected given the general effect of selection on linked neutral sites [68]. The fact that our estimates of both effective gene flow and effective population sizes correlate with recombination in the expected direction suggests that we have successfully decomposed the effects of selection acting at barrier loci from BGS and positive selection.

The fraction of chromosomes inferred to be covered by strong barrier regions (i.e. $\Delta_{B,0} > 0$) varies widely between chromosomes. For instance, chromosomes 1, 4, 9, 14 and 20 harbour no $\Delta_{B,0}$ barrier windows at all, while chromosome 18 appears to have 6% of its physical length

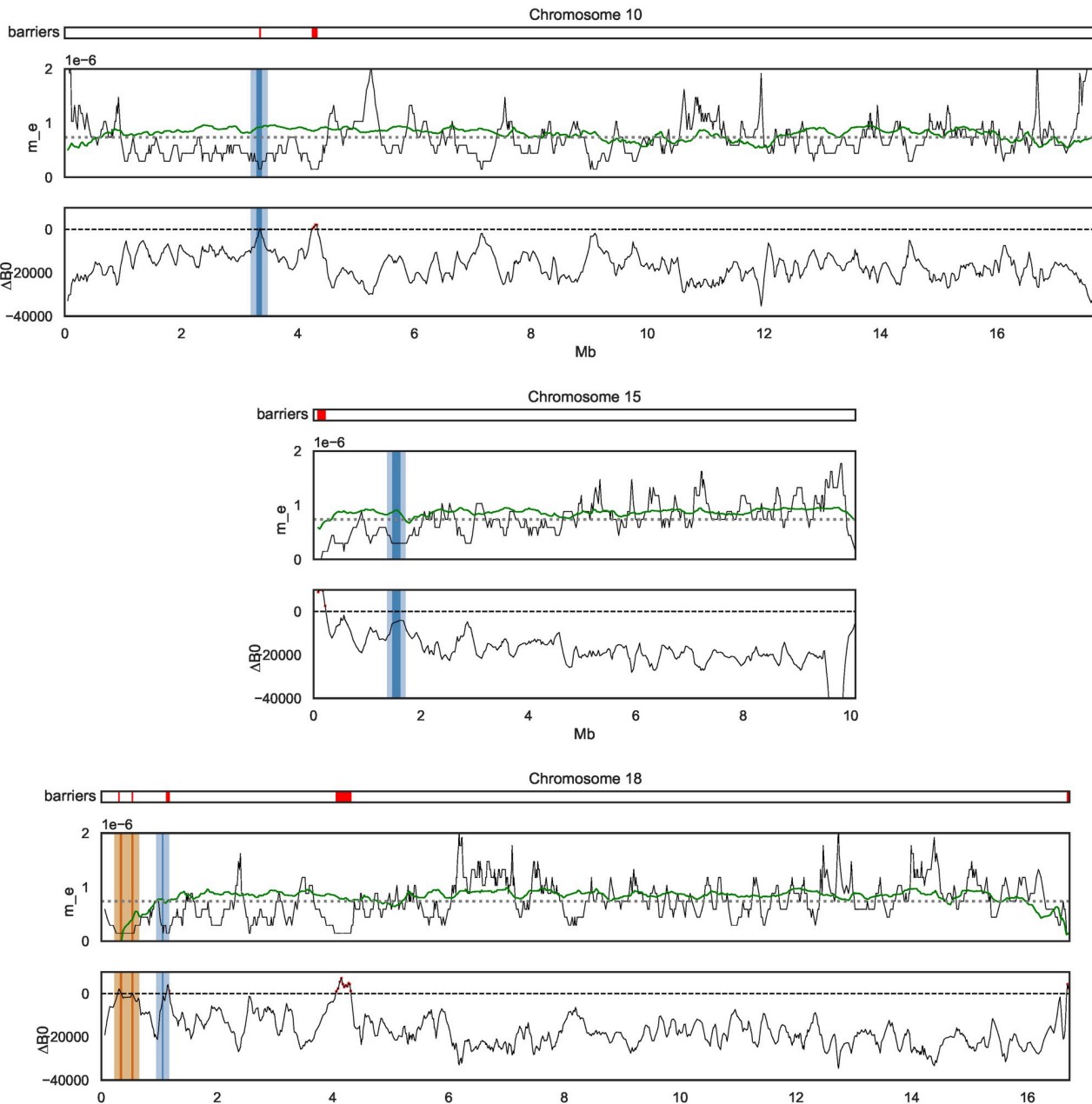

**Fig 5. Barrier windows inferred using `gIMble` include known large effect loci for wing pattern traits.** Local point estimates of $m_e$ (top of each subplot) and $\Delta_{B,0}$ (bottom of each subplot) for the three chromosomes containing large-effect wing patterning genes (shown in blue). $\Delta_{B,0}$ identifies *wnt-A* [59] on chromosomes 10 (top) and *optix* on chromosome 18 (bottom, blue) as barriers. Given that causal sites may be situated in regulatory regions, which in the case of *optix* extend to $\sim$ 100 kb away from the gene [60], we have highlighted the 100 kb up and downstream of each gene (lighter blue). $\Delta_{B,0} > 0$ barriers for each chromosomes are shown in red on top. Another $\Delta_{B,0} > 0$ barrier on chromosome 18 coincides with *Grik2* and *regucalcin2* (orange), genes that are associated with male wing-pattern preference. The wing-pattern gene *cortex* on chromosome 15 (centre) is not a barrier according to $\Delta_{B,0}$. We fit a polygenic model (modified from [9]) that predicts local $m_e$ (shown in green) as a function of CDS density and recombination rate.

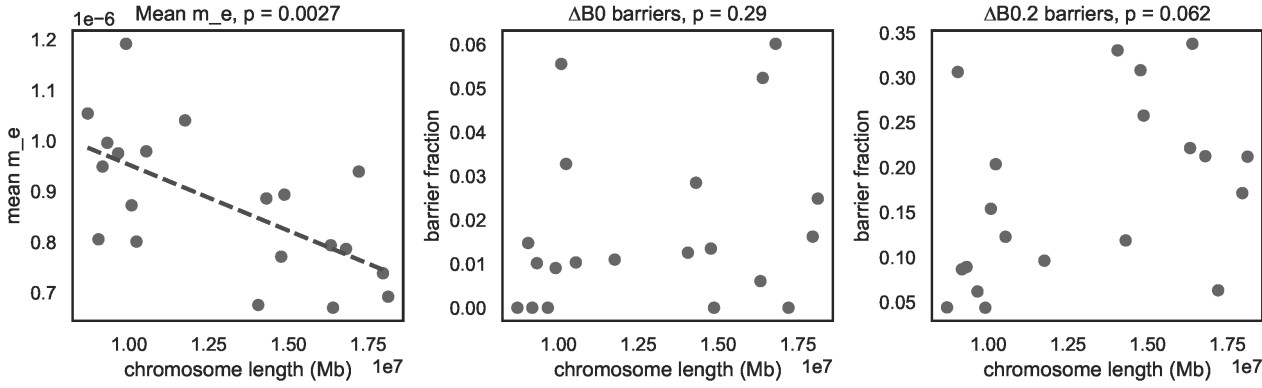

**Fig 6. Relationship between chromosome length and the strength and abundance of barriers.** Barrier strength, measured as the mean $m_e$ across windows, is negatively correlated with chromosome length (left). In contrast, there is no significant relationship between the relative length of chromosomes defined as barrier regions and chromosome length irrespective of whether barriers are defined via $\Delta_{B,0}$ (center) or $\Delta_{B,0.2}$ (right).

covered by strong barriers. However, the fraction of chromosomes covered by barrier regions is uncorrelated with physical chromosome length (Fig 6, centre and bottom). This is interesting for two reasons. First, under the simplifying assumptions that each barrier region is caused by a single barrier locus and that selection acting on individual barrier loci does not differ among chromosomes, the width of barriers is predicted to scale with $1/r_{bp}$ [4]. If the rate at which mutations give rise to potential barrier alleles is uniform across chromosomes, we would expect longer chromosomes to have a greater concentration of barrier windows. Second, theory also predicts that existing barrier loci facilitate the establishment of further barrier loci in close linkage [63, 64]. This clustering effect would again be stronger for long chromosomes (lower $r_{bp}$) compared to short chromosomes (higher $r_{bp}$), and should thus further contribute to a positive correlation between chromosome length and the fraction of chromosomes covered by barrier regions. The most likely explanation for the absence of this predicted correlation for $\Delta_{B,0} > 0$ barrier regions is a lack of statistical power (i.e. there are only 25 $\Delta_{B,0}$ barrier regions in total). Indeed, when we consider the less-stringent barrier definition (i.e. the 172 barrier regions identified by $\Delta_{B,0.2} > 0$), a weakly positive, albeit non-significant, positive relation between chromosome length and the barrier fraction emerges. (Fig 6, right).

At the second, intra-chromosomal scale, we find that window-wise estimates of $m_e$ are strongly and positively correlated with $r_{bp}$ [67] (estimated directly from lab crosses, see Methods), again as predicted under a polygenic architecture of barriers [69]. Similarly, barriers to gene flow defined either by $\Delta_{B,0}$ or $\Delta_{B,0.2}$ are concentrated in regions of reduced recombination: as per a simple resampling procedure, $r_{bp}$ is significantly lower in barrier windows compared to the average (0.75 vs 1.94 cM/Mb, respectively) (Fig 7 and Table 4). Assuming that selective targets (both of BGS and selection on barrier loci) reside in or near coding DNA sequence (CDS), we also expect a negative relationship between the density of CDS and local $N_e$ and $m_e$ estimates. We find clear support for both relationships (Fig I in S1 Text, e.g. for $m_e$ and CDS measured at a 250 kb scale Pearson's $\rho = -0.164$, $p < 0.000001$). Thus, overall, and in agreement with a model of polygenic selection against inter-specific gene flow, we find that barriers to gene flow are concentrated in regions of low recombination and high CDS density.

## How many barrier loci are there?

It may be tempting to simply interpret the count of barrier regions (Table 4) as an estimate of the number of barrier loci. However, this is naive for several reasons: firstly, gIMble models

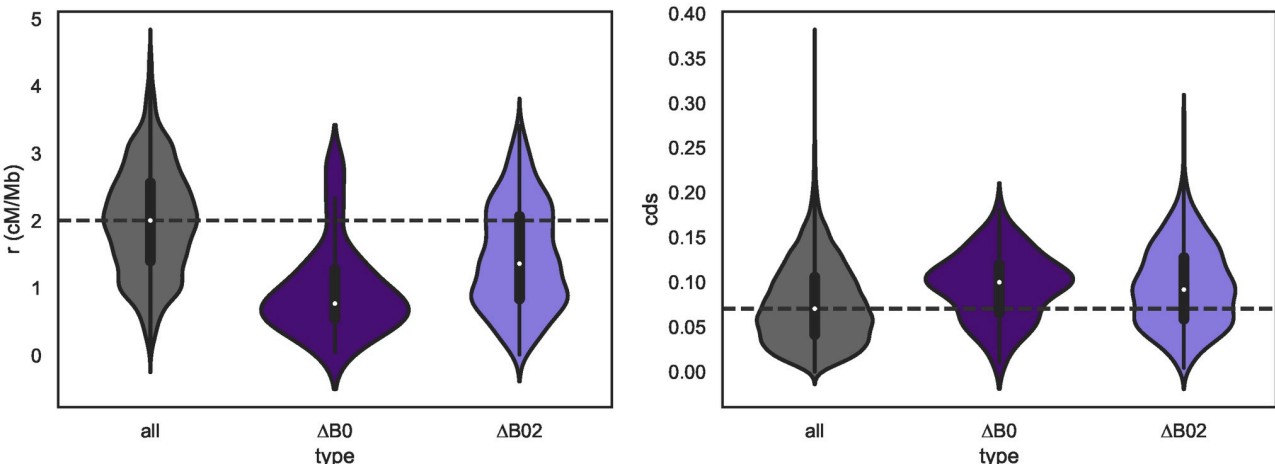

**Fig 7. Barrier windows show reduced recombination and an enrichment for coding sequence.** Left: genomic windows diagnosed as barriers to gene flow ($\Delta_{B,0.2}$ in light indigo, $\Delta_{B0}$ in dark indigo) have reduced recombination rate. The median $r$ (cM/Mb) across all windows is shown as a dashed line. Right: Barrier windows are enriched for coding sequence (CDS).

the indirect and aggregate effects of barrier loci on neutral sequence, and although we expect strong barriers to lie within barrier regions identified by `gIMble`, there is no direct correspondence between the number of causal loci and the number of barrier regions, especially if barriers are polygenic. Secondly, the power to detect barriers relies both on the background rate of gene flow and the $m_e$ threshold used to define barriers (although under the IM model $m_e* = 0$ gives a non-arbitrary definition of complete/strong barriers). Thus an alternative approach to get at the architecture of barriers is to interpret the variation in $m_e$ inferred by `gIMble` in light of a polygenic model.

Aeschbacher et al. [70] proposed a framework to quantify the strength of selection against introgression under a simple model in which barrier loci are uniformly distributed across the genome with density $\nu$ per base pair. Under this model, the expected local reduction in $m_e$ at a genomic window $i$ relative to the background rate (i.e. $m_{e,i}/m$) is a function solely of the per-base pair recombination rate ($r_{bp}$) and the "selection density" $\sigma = \nu s$, where $s$ is the selection coefficient at each barrier locus [70, eqn. 1.9 in Appendix 1]. We fitted a version of this model in which the reduction in gene flow at a focal genomic window depends only on the closest flanking barrier locus on either side of it. To account for variation in CDS as a proxy for the density of barrier loci, we modified this model by scaling $\nu$ by the inverse of the CDS density. We inferred $\sigma$ and the background $m$ by minimising the sum of squared differences between the expected local $m_e$ (given $r_{bp}$ and CDS) and the $m_e$ estimates inferred by `gIMble` across all genomic windows. Exploring CDS and $r_{bp}$ measured at a range of physical scales around each window (see Methods) revealed that $r_{bp}$ estimated at a scale of 2 Mb and CDS measured at a 250 kb scale best explained the variation in inferred $m_e$ (Fig J in S1 Text). However, given the

**Table 4. Average properties and $N_e$ estimates for genomic windows defined as barriers between *H. melpomene* and *H. cydno*.**

| Partition | # of windows | # of regions | Total (Mb) | %CDS | r(cM/Mb) | $N_{cyd}$ | $N_{mel}$ |
|---|---|---|---|---|---|---|---|
| $\Delta_{B,0} > 0$ | 87 | 25 | 4.9 | 0.0958 | 0.956 | 1,022,000 | 430,000 |
| $\Delta_{B,0.2} > 0$ | 871 | 170 | 47.4 | 0.0956 | 1.451 | 1,283,000 | 513,000 |
| autosomes | 11, 217 | n/a | 259 | 0.0765 | 2.004 | 1,506,000 | 630,000 |

edge effects accumulating at chromosome ends when measuring $r_{bp}$ over 2 Mb, especially for the shorter chromosomes with a length on the order of 10 Mb (Fig I in S1 Text), we chose to use $r_{bp}$ values estimated over 1 Mb instead of 2 Mb. With this combination of ranges (1 Mb for $r_{bp}$, 250 kb for CDS), we obtain estimates of $\hat{\sigma} \approx 7.77 \times 10^{-9}$ (95% CI: [$6.95 \times 10^{-9}$, $8.59 \times 10^{-9}$]) and of the background migration $\hat{m} \approx 9.87 \times 10^{-7}$.

The compound nature of $\sigma$ means that similar values can be produced by a small number of barrier loci under strong selection and a large number of barrier loci under weaker selection. One question we can ask is what strength of selection $s$ per barrier locus would be required to produce our estimate of $\sigma$, assuming the signal was caused only by the 25 strong (i.e. $\Delta_{B,0}$) barrier regions we inferred. For the sake of simplicity, we assume that each barrier locus experienced the same selection, and that each barrier region contains exactly one barrier locus. Given a total length of 259 Mb for *H. melpomene* autosomes [54], we find $\hat{s} = \hat{\sigma}/(25/[259 \times 10^{6}]) = 0.0805$, which would imply extremely strong selection at each individual barrier locus. However, the estimate of $\sigma$ is clearly driven by the genome-wide relationship between $m_e$ and $r_{bp}$ (and CDS). Indeed, exclusion of the 25 $\Delta_{B,0}$ outlier regions has almost no effect on our estimate of $\sigma$ and $m$: we estimate $\hat{\sigma} \approx 7.07 \times 10^{-9}$ (compared to $\hat{\sigma} \approx 7.77 \times 10^{-9}$) and $\hat{m} \approx 9.82 \times 10^{-7}$ (compared to $\hat{m} \approx 9.87 \times 10^{-7}$). We therefore conclude that there must be many more barrier loci than $\Delta_{B,0}$ regions. We can obtain an upper bound for the number of barrier loci if we consider that $2N_e s > 1$ for each barrier locus for selection to be stronger than genetic drift. At the same time, selection at each barrier locus has to be stronger than the background migration rate if the barrier is to be maintained (i.e. $s > m$) [3]. Given our estimates of $\hat{N}_{mel} \approx 549,000$ (Table 2) and $\hat{m} \approx 9.87 \times 10^{-7}$, the former threshold ($2N_e s > 1$) is the limiting one, and the maximum number of barrier loci implied by $\hat{\sigma} = 7.77 \times 10^{-9}$ is $\approx 1,100,000$. Thus, given this upper limit for the number of barrier loci, we cannot rule out that the overall species barrier between *H. melpomene* and *H. cydno* has an omnigenic architecture. In other words, while a moderate number of strong barriers—including those associated with the candidate genes *optix*, *regucalcin2*, and *wnt-A*—appear to locally reduce $m_e$ to a large extent, a much larger number of weaker barrier loci likely contribute to reproductive isolation between these taxa.

## Robustness to model choice

Our reconstruction of $m_e$ assumes an *IM* model, while our analysis of how polygenic selection generates this $m_e$ variation assumes a model of migration–selection balance (i.e. genetic variation at barrier loci is maintained at migration–selection equilibrium). Interestingly, the second best global model to the *Heliconius* data is a history of migration with infinite split time (*MIG*, $T \to \infty$). This and the fact that global parameter estimates of $m$, $N_{cyd}$ and $N_{mel}$ under the *MIG* model are very similar to those under the *IM* model (Table B in S1 Text) implies that the information about both $T$ and the ancestral $N_e$ are limited. This raises the question whether these two parameters matter at all for our inference of $m_e$ heterogeneity and barriers.

To assess the extent to which the *Heliconius* barrier analysis is model dependent, we set up a simpler gIMble scan under the *MIG* model (varying $N_{cyd}$, $N_{mel}$ and $m_e$). While $\Delta_{B,0}$ barriers are undefined under the *MIG* model, we find that estimates of both $m_e$ variation and $\Delta_{B,0.2}$ barriers are extremely similar to the analogous results under the *IM* model (Fig K in S1 Text). Thus while we would argue that conceptually (and in general) a two-step inference procedure is a logical approach, our results suggest that for *H. melpomene* and *H. cydno* the first step does not matter, i.e. the inference of barriers in this dataset does not depend on an inferred global $T$.

## Discussion

Progress in speciation genomics has been impeded by a conceptual and methodological divide between approaches that model speciation history under neutral scenarios and analyses that aim to identify loci under divergent selection that act as barriers to gene flow. Here, we bridge this divide by introducing gIMble: a single framework to model both the overall species history and the genomic landscape of speciation. We used gIMble to demonstrate a two-step inference procedure that fits a genome-wide (global) demography under the IM model, and then relaxes constraints on $N_e$ and $m_e$ to estimate these two parameters locally in sliding windows along the genome. Unlike previous approaches that attempt to describe species barriers via summary statistics such as $F_{ST}$ which is affected by all forms of selection, the effective demographic parameters inferred by gIMble have straightforward interpretations in terms of evolutionary processes: while heterogeneity in $N_e$ largely reflects BGS, $m_i$ measures the long-term combined effects of barriers to gene flow at a local genomic window $i$ [61]. The ability of our framework to identify both the position and strength of barriers to gene flow based on estimates of $m_e$ holds promise for comparative studies of species barriers [35].

Applying our inference procedure to the neotropical butterfly species *H. cydno* and *H. melpomene*, we find strong support for genome-wide variation in $N_e$ and $m_e$. Genomic windows of significantly reduced $m_e$ partially overlap with well-known large-effect loci controlling wing patterning (*optix*, *cortex* and *wnt-A*) and male preference for wing pattern (*regucalcin2*). We also confirm previous studies that estimated higher (effective) gene flow from *H. cydno* into *H. melpomene* than in the opposite direction [65, 71]. Finally, we illustrate how the inferred genome-wide variation in $m_e$ can be used to infer the aggregate strength of selection against gene flow given maps of CDS density and recombination rate. This analysis shows that the barrier between these species has a substantial polygenic component.

The inference framework we have developed is a first step towards model-based characterisation of species barriers. However, there is extensive room to improve our method both in terms of model complexity and optimal exploitation of genomic data. It is important to reiterate that modelling the effect of barriers via demographic parameters is a drastic simplification [72]. While the approximation implemented in gIMble does capture the effect of selection *maintaining* barriers once they have been established, it does not consider the processes by which alleles at barrier loci (and locally beneficial alleles more generally) become established in the first place. While Bateson–Dobzhansky–Muller incompatibilities can arise through drift alone only in the absence of migration [73], their establishment in the face of gene flow involves selection favouring locally adaptive alleles. Thus, developing methods that capture the joint effect of selection establishing barriers and the resulting reduction in $m_e$ remains an important goal of future work.

While gIMble is focused on subsamples of pairs of diploid genomes, its modular design facilitates the implementation of generalisations to larger samples and more complex demographic models. There are several obvious and useful avenues for future extensions. First, gIMble assumes a constant rate of unidirectional gene flow that persists until the present. Since many taxon pairs that are of interest in speciation research are likely to have completed speciation (i.e. there is no contemporary gene flow), it would be useful to generalise the method to allow for a time at which gene flow ceases—the Isolation with Initial Migration (IIM) model [74]—as well as bi-directional gene flow. While an approximate solution to the problem of bi-directional gene flow has been described [71], we currently do not know of a closed-form analytic solution for a sample of four lineages that accounts for multiple discrete demographic events in the IIM model as is required by agemo [42] (see Methods). While there are reasons to expect strong asymmetry in the direction of gene flow in general [75], it

would be useful to know how sensitive the strictly unidirectional `gIMble` scans are to a small amount of counter-directional gene flow. Much of the information about past gene flow is contained in the frequency of pair-blocks carrying shared heterozygous sites that are generated when migrant lineages coalesce in their source population (Fig 1A). We therefore expect the $m_e$ estimate to be inflated by migration in the direction opposite to that of the best-fitting model of unidirectional gene flow.

A perhaps more important question is to what extent `gIMble` analyses are meaningful when the true underlying history involves secondary contact rather than continuous gene flow. In the absence of extensions that allow fitting multiple discrete events, e.g. models of instantaneous admixture or periods of gene flow, we caution the user to first assess the overall fit of the IM model using other inference approaches to determine if a `gIMble` analysis is appropriate for their dataset. In the case of the focal *Heliconius* pair, we not only have a strong prior for continuous gene flow but can also show that the `gIMble` scan is robust to the choice of model (*MIG* vs *IM*).

Second, we have ignored heterogeneity in the mutation rate ($\mu$) in our analyses which may be confounded with heterogeneity in $m_e$. However, there is no direct evidence from mutation accumulation experiments [76] for heterogeneity in $\mu$ over the scale relevant for our analysis, i.e. window of 100kb. In other words, it is biologically implausible that the $m_e$ variation we have inferred reflects mutation rate heterogeneity. In contrast, $\mu$ is likely to vary over the scale of `gIMble` blocks. Moreover, other aspects of genome data (mis-annotation, mapping errors) are likely to affect the window-wise bSFS in similar ways to fine-scale in $\mu$. Such fine-scale variation in $\mu$ should lead to an overall upward bias in estimates of $N_e$ and $m_e$ (which is conservative when identifying barriers).

Finally, `gIMble` currently does not make use of phased sequence data. While this feature makes it useful for studies where phasing is not achievable (e.g. short-read sequencing data and small sample sizes as considered here), extending `gIMble` to phased data would increase its statistical power. A further and more profound extension would be to incorporate haplotype information in the "blocking" algorithm. Currently, we blindly discretise the genome into blocks of a fixed length. Given that the history of each block is modelled assuming a single genealogy, blocks should ideally be defined via topologically informative sites (e.g. using the four-gamete-test and other topology information) to ensure that each block meets the assumption of a single genealogy [77]. Recently, progress has been made on approaches that infer ancestral recombination graphs (ARGs) from phased data. In particular, several minimal-assumption heuristics exist to infer topology-only ARGs [78–80]. These methods use information from neighbouring haplotypes to reconstruct marginal tree topologies and would allow blocks to be defined by the spans of marginal trees, rather than discretizing the genome at random and assuming no recombination within blocks, as we have done here. Thus, developing sampling strategies and likelihood calculations that incorporate topology and haplotype information will be a priority of future work.

Our analysis of *Heliconius* data illustrates the potential of our framework for speciation research. Specifically, we show that the $m_e$ variation inferred by `gIMble` can be used to answer two different, yet complementary, sets of questions about the speciation process. On the one hand, one may assume that reproductive isolation is conferred at least in part by a small to moderate number of large-effect barrier loci [1] that can be identified individually from their strong effect on $m_e$. A scan based on $\Delta_{B,0}$ yields a meaningful estimate of the fraction of genome in this category. Importantly, our approach—unlike outlier scans based on summary statistics such as $F_{ST}$—avoids arbitrary significance thresholds. Instead, the power to estimate barriers to gene flow (which can be quantified using `gIMble simulate`) is a function of the background demography (e.g. the higher the genome-wide background $m_e$, the

greater the power to identify $\Delta_{B,0}$ barriers) and the recombination rate. On the other hand, reproductive isolation may be largely, if not entirely, polygenic, with a large number of weak barrier loci reducing gene flow genome-wide [61]. In this case, it becomes impossible to identify individual barriers [81]. Instead, the main objective will be to understand the genomic predictors/correlates of $m_e$. One advantage of gIMble is that its inference of $m_e$ and $N_e$ is purely based on genomic variation and can be intersected with many other data types that are relevant for speciation, e.g. gene expression data or information about barriers to gene flow from crossing experiments or hybrid zone studies.

Importantly, gIMble also does not rely on recombination information (we assume that blocks are non-recombining and statistically exchangeable), which means that the interaction between recombination rate and $m_e$ can be modelled in a second step to make an aggregate inference on the nature of the polygenic species barrier. We have provided an example of such an inference using the framework of Aeschbacher et al. [9], extended here to account for variation in CDS density as a proxy for the density of candidate barrier loci.

Combining these two approaches, we find that *H. melpomene* and *H. cyndo* are separated by a mixture of large-effect barrier loci and polygenic barrier. Three out of four known large-effect loci controlling barrier phenotypes in this *Heliconius* species pair are contained in the 2% of the genome identified as a barrier by the strictest definition of no effective migration, i.e. $\Delta_{B,0} > 0$. Yet at the same time, it is clear that $\Delta_{B,0}$ barriers do not drive the genome-wide correlation we find between $m_e$ and $r_{\text{bp}}$ and which is little affected by the exclusion of $\Delta_{B,0}$ barriers. Simple scaling arguments lead us to conclude that, apart from the 25 major barrier loci we identified, up to hundreds of thousands of weak barrier loci scattered across the genome might contribute to reproductive isolation between *H. melpomene* and *H. cyndo*. In fact, the large-effect wing pattern loci may be a rather unique feature of the strong selection pressures generated by Müllerian mimicry in *Heliconius* and it is unlikely that they function as the sole, or even the truly significant, species barriers between *H. melpomene* and *H. cydno* for several reasons. First, these species are known to differ in numerous other ecological traits, including habitat preference [82]. Second, *H. melpomene* alleles at both *optix* and *cortex* are shared between taxa [83, 84] and are known to have adaptively introgressed into *H. timareta* (a relative of *H. cydno*), yet the species barrier between *H. melpomene* and *H. timareta* remains intact [85, 86]. Finally, only three of the 25 strong barrier regions we identify map to genes that are known to control wing patterns or preference for wing patterns.

Given that much of the past progress in speciation research has come from comparative analyses [28, 87], the fact that even the relatively simple model-based scans we have implemented here open the door to comparative studies of species barriers is cause for optimism. In particular, we argue that systematic comparisons of demographically explicit genome scans across many taxon pairs will reveal whether the combination of a few strong barrier regions embedded in a genome-wide, highly polygenic barrier signal seen in *Heliconius* is the rule or an exception.

## Materials and methods

### Implementation

The gIMble analysis workflow is designed for WGS data in which all samples have been mapped to a single reference and largely uses established bioinformatic pipelines and file formats. Fig A in S1 Text shows the basic workflow we have used to analyse the *Heliconius* data. Currently, gIMble only supports analyses of data from population/species pairs.

The different steps of the workflow are implemented as distinct modules, allowing users to tailor the analysis steps to a particular dataset and set of research questions:

- `gimbleprep` applies simple coverage and quality filters to the input files to ensure consistency in how variation is sampled along the genome.

- `parse` reads the pre-processed input files into a compressed (`zarr`) datastore.

- `blocks` partitions genomic variation data into pair-blocks composed of one individual sampled from each population.

- `windows` creates (sliding) windows of a fixed number of pair-blocks along the genome.

- `info` displays information about the output data of `blocks` and `windows`.

- `tally` prepares `blocks`/`windows` data for the `optimize` and `gridsearch` modules.

- `optimize` performs parameter optimization given a bSFS tally and a bounded parameter space under a specified model.

- `makegrid` pre-computes and stores the probabilities of bSFS configurations for a grid in parameter space.

- `gridsearch` computes and stores likelihoods of `tally` data given a pre-computed `makegrid`.

- `query` writes output files describing `windows`, `tally`, `optimize`, and `gridsearch` results.

- `simulate` uses `msprime` [43] to simulate tallies which can be used for parametric bootstrapping and/or power analyses.

Below, we briefly describe the pre-processing of data, the organisation of `gIMble` analyses into a `zarr` datastore, the modules that allow the fitting of global and local models, and the parametric bootstrapping of `gIMble` estimates.

### Data processing

**`gimbleprep`.** The `gimbleprep` module implements a simple and consistent filtering of input files for subsequent analyses. The purpose of this module is to standardise the quality of data analysed by `gIMble` using a minimal set of filters that apply to both variant and invariant sites and which maximise the proportion of the data retained for analysis.

The required input for `gimbleprep` consists of the reference genome (FASTA format), read sets of each sample aligned to the reference (BAM format), and the called variants (VCF format). The output is composed of the following four files:

- **genome file** records the name and length of each sequence in the FASTA file.

- **sample file** specifies the name of each sample in the VCF file. This file needs to be manually edited by the user in order to partition samples into two populations/species.

- **VCF file** contains only those variants that passed the filter criteria: a minimum distance from non-SNP variants, adequate read depth (based on min/max thresholds inferred from BAM files), minimum genotype quality, and minimum read mapping quality (SAF, SAR, RPR, RPL).

- **BED file** describes the "callable" genomic regions for which each sample has adequate coverage (based on min/max thresholds). Only those regions described here will be visible to `gIMble`. The user can process this file further using external tools to exclude certain genomic regions, such as genes and repeats.

The genome and sample files give the user control over what data are included in subsequent analysis steps of `gIMble`: the genome file delimits the parts of the genome to analyse and the sample file specifies the samples (and their population membership) from which block-wise data is tabulated.

The VCF and BED files generated by `gIMble preprocess` describe the genomic variation across samples that satisfy the previously-defined quality thresholds and therefore serve as a starting point for any complementary or alternative analyses outside of `gIMble` that the user may wish to run on the same data.

**gIMble parse.** This module parses the four input files generated by `gimbleprep`. Based on the parsed data, a compressed `zarr` datastore is created which is central to all subsequent analyses.

**gIMble blocks.** The module `blocks` uses the "callable" regions specified in the BED file and the variants contained in the VCF file to define pair-blocks of a fixed number of callable sites. Defined this way, pair-blocks can also span non-callable sequence and therefore may have variable physical spans in the genome. To maximise the genomic regions sampled and to avoid biases due to missing data, blocks are constructed independently for each sample pair (hence our term pair-blocks). This ameliorates the asymmetry in coverage profiles among the samples due to stochastic variation in sequencing depth and/or reference bias. If, instead, blocks were constructed only for sites that pass the filtering criteria among *all* samples, the "block-able" fraction of genome would become smaller and more biased as sample size increases.

Although blocks are constructed for all pairwise sample combinations both between the two populations (X) and within each population (A and B), only heterospecific pair-blocks (X) are used in the inference steps. The blocking of genomic data is controlled by the parameters `--block_length`—the number of callable sites in each pair-block—and `--block_span`—the maximum distance between the first and last site in a pair-block. While `gIMble` only considers biallelic genotypes for inference, the block-cutting algorithm allows for multiallelic and missing genotypes. The number of each permitted in a pair-block is controlled with the parameters `--max_multiallelic`, and `--max_missing`. Note that only one set of blocks can be stored in a `gIMble` datastore.

**gIMble windows.** Windows are constructed by traversing each sequence of the reference from start to end, incorporating the heterospecific pair-blocks (X) as they appear (based on their start positions). The parameter `--blocks` controls the window-size, i.e. how many pair-blocks are incorporated into each window and the parameter `--steps` by how many blocks the next window is shifted. Since differences in coverage or mapping bias among samples can cause uneven distributions of blocks along the genome, there is no guarantee that all heterospecific sample pairs contribute an equal number of pair-blocks to a window. However, by choosing a large enough value for `--blocks` this problem can be mitigated. Analogous to blocks, only one set of windows can be stored in a `gIMble` datastore at a time.

**gIMble info.** `gIMble info` computes standard population genetic summary statistics ($\pi$, $d_{xy}$ and $H$ mean heterozygosity) using the pair-blocks sampled both between (X) and within species/populations (A and B) that are recorded in the `gIMble` datastore.

**gIMble query.** Data contained in the `gIMble` datastore can be queried using this module and written into BED and TSV files. This allows extraction of the results of the modules `windows`, `tally`, `optimize`, `makegrid`, `simulate`, and `gridsearch`.

**gIMble tally.** Having created blocks (with `gIMble blocks`) and windows (with `gIMble windows`), the module `gIMble tally` generates the bSFS tally for heterospecific sample pairs, each consisting of a single diploid sample from each population/species A and B. The bSFS tally [71] of a dataset or window enumerates the bSFS configurations which are

themselves indexed by vectors of the form $\underline{k}_i$, i.e. counts of the four possible mutation types i = {het_b, het_a, het_ab, fixed_diff} found within a pair-block:

- het_b sites with homozygous genotype in A, heterozygous genotype in B

- het_a sites with heterozygous genotype in A, homozygous genotype in B

- het_ab sites with shared heterozygous genotypes in both populations

- fixed_diff sites with distinct homozygous genotype in both populations

The argument --kmax limits the counts of mutations of each type in the bSFS. Counts of mutations inside blocks ({het_b, het_a, het_ab, fixed_diff}) exceeding the values in --kmax (default: 2, 2, 2, 2) are combined. Thus the choice of --kmax involves a trade-off between the information gained by distinguishing the probabilities of rare bSFS configurations and the increase in computational complexity. The bSFS tallies of heterospecific data can be counted from all pair-blocks (-d blocks, i.e. genome-wide), per window (-d windows), or for all pair-blocks that are included in windows (-d windowsum, in this case pair-blocks included in multiple overlapping windows are counted repeatedly).

## Calculating likelihoods

Probabilities of block-wise data under the coalescent with arbitrary demographic history can be calculated using the method described in [40, 41]. In this framework, the generating function (GF) for the distribution of genealogical branch lengths is described recursively as the convolution of independent Laplace-transformed exponential random variables. The GF for a particular model and sampling configuration can be represented most simply as a directed graph involving all possible ancestral states of the sample, and the probability of any bSFS configuration $\underline{k}_i$ can be calculated as a partial derivative of the GF [41].

Previous automation of this calculation relied on rather inefficient use of *Mathematica*'s [88] symbolic algebra functionality. gIMble now uses agemo [42], a user-friendly re-implementation of the GF approach in python that extracts the probabilities of bSFS configurations efficiently by means of a graph traversal algorithm.

gIMble accommodates large samples by considering the composite likelihood (CL) across all $n_A \times n_B$ heterospecific sample pairs (each pair $j$ consists of single diploid individual, one each from $A$ and $B$). Using $n_{i,j}$ as shorthand notation for the number of blocks in sample pair $j$ with bSFS configuration $\underline{k}_i$, the joint log composite likelihood (ln $CL$) is given by:

$$\ln CL(\Theta) = \sum_i \sum_{j=1}^{n_A \times n_B} n_{i,j} \ln p(\underline{k}_i | \Theta) \tag{2}$$

where $\Theta$ is the vector of model parameters. The composite likelihood can be calculated explicitly for each point in parameter space that is visited during the optimization run for a given data set (see optimize). However, when scanning WGS data, it is more efficient to cast a grid over parameter space and pre-compute ln $p(\underline{k}_i | \Theta_h)$ for all $\underline{k}_i$ and each possible possible parameter combination $h$ in the grid (see makegrid). Composite likelihoods can then be computed efficiently for tallies (based on blocks, windows, or simulate) using gridsearch.

**gIMble optimize.** This module infers the best-fitting parameters under a given model for a given tally (based on parsed or simulated data) using an optimization algorithm with bounded constraints (CSR2, sbplx, or neldermead), as implemented in nlopt [89]. Given a set of bounds, optimization can be initiated either at the midpoint of the bounded parameter space

or at a random starting point. Optimizations finalise after user-defined stopping criteria are met. The user can assess convergence of the optimizer by consulting the log file.

**gIMble makegrid and gridsearch.** Precomputing the probabilities of bSFS configurations in a grid across parameter space is efficient (relative to using an optimization algorithm) and therefore useful when we wish to interrogate data in replicate, i.e. across windows or simulation replicates. Grids are pre-computed using makegrid, saved in the gIMble store, and used to analyse any dataset contained within the same datastore using gridsearch. A grid search may be used either for an initial exploration of the likelihood surface (i.e. prior to defining the parameter space across which to run *optimize*), or to fit models to window-wise or simulation data.

## Parametric bootstrap

Maximizing ln *CL* across blocks and sample pairs ignores both linkage between blocks and the pseudo-replication inherent in subsampling heterospecific pairs. We therefore implemented a wrapper for msprime [43] to facilitate quantification of uncertainty in parameter estimates, obtaining critical values of Δ ln *CL* (for model comparisons) and measuring the power of $\Delta_B$ scans via parametric bootstrap.

**gIMble simulate.** This module simulates replicate datasets either under a user specified IM type model and set of parameter values or based on parameters inferred across windows from a previous gridsearch. Each simulation replicate consists of a set of windows, and so can be simulated to correspond to a real dataset in terms of the number of windows and their recombination rates. Simulated data can be analysed in the same way as real data. This allows users to perform parametric bootstraps on both global (genome-wide) and local (window-wise) estimates and to quantify the power and potential biases (due to model misspecification) of gIMble analyses, in particular the assumption of no recombination within blocks. Parametric bootstraps on local estimates are set up by providing the key of the corresponding gridsearch analysis.

The module *simulate* uses demes [90] internally for unequivocal model specification. gIMble can also output the results of optimize as a demes-readable yaml-file, allowing users to connect gIMble to the growing list of other population genetics tools that support the demes format [43, 91–95].

## Heliconius analyses

We provide a list of all gIMble commands used to analyse the *Heliconius* dataset in the Supplementary Methods in S1 Text.

**Sequence data.** Raw reads (Illumina 150 base paired-end) (Table A in S1 Text) were downloaded from ENA and quality trimmed using fastp 0.20.1 [96] (--length_ required 50). Trimmed reads of each read set were aligned against the chromosome-level assembly of *H. melpomene* Hmel2.5 [67] using bwa mem 0.7.17-r1188 [97]. Alignment files were sorted with samtools 1.12 [98] and duplicated read pairs were marked with sambamba markdup 0.8.1 [99]. Variants were called using freebayes v1.3.3 [100] (-k -w -j -T 0.02 -E -1 --report-genotype-likelihood-max).

**Data processing.** We used the following gimbleprep parameters for the *Heliconius* analysis: --min_qual 1, --snpgap 2, --min_depth 8 (i.e. 8 reads), and --max_ depth 3 (i.e. 3 × mean coverage of each BAM file). Blocks were cut using the following parameters: --block_length 64, --block_span 128, --max_multiallelic 3, and --max_missing 3. Blocks were grouped into windows using gIMble windows with these parameters: --blocks 50000 (on average, each of the 100 heterospecific sample

pairs should occur in 50,000/100 = 500 blocks within a window) and --steps 10000. Tallies were made using gIMble tally with --kmax 2,2,2,2 for 'blocks', 'windows', and 'windowsum' (all blocks in windows).

**Recombination map and parametric bootstrap.** For each data window in the *Heliconius* analysis, we obtained direct recombination rate estimates $r_{bp}$ over a range of scales from the total number of crossover events recorded by Davey et al [67]: we averaged the recombination rate estimates in *H. melpomene*, *H. cydno*, and hybrid crosses for a fixed physical region (of size 125 kb, 250 kb, 500 kb, 1 Mb, 2 Mb) centred on the window mid-point using a custom script get_recombination_rate_KL.py.

We performed a parametric bootstrap on the global *IM* estimates obtained for *Heliconius*. To obtain 95% confidence intervals (CI) of MCLEs of parameters under the best fitting global *IM* model (Table 2), we simulated 100 replicate datasets under the inferred $IM_{\rightarrow mel}$ model. Each replicate consisted of a dataset partitioned into windows analogous to the real data (i.e. 500 blocks of 64 bases sampled across $10 \times 10$ heterospecific pairs). Importantly, simulations were conditioned on the *H. melpomene* recombination rates ($r_{bp}$ measured at a 1 Mb scale). Given that real data windows partially overlap and are in LD, whereas simulated windows are assumed to be statistically independent, we simulated datasets containing 50 times fewer windows than the real data (this ensures a distance of around 500 kb between windows). We estimated the 95% CIs of data MCLEs as +/- 2SD of the MCLEs across replicates.

To estimate the uncertainty in the gIMble scan, we quantified the false positive rate (FPR) as follows: For each window and its associated $r_{bp}$ (measured at a 1 Mb scale), we simulated 100 replicates under the locally best fitting set of **N** estimates but fixing $m_e$ to $\hat{m}_e$ and $\hat{T}$ to the MCLE under the global model (Table 2). We used gIMble gridsearch to obtain MCLEs of $m_e$ for each replicate and measure the FPR as the fraction of the simulation replicates for which $\Delta_B > 0$.

## Power analysis

While the choice of block length involves a trade-off between power and bias, the definition of windows involves a trade-off between power and resolution. Our analysis scanned the *Heliconius* genomes for barriers using windows on the scale of 100 kb (i.e. $50,000 \times 64$ base blocks), but we find that given the *Heliconius* background demography, there is high power to detect genomic barriers for a wide range of window sizes (Fig L in S1 Text).

We quantified the impact of window size, recombination rate and sample size on the power to detect barriers using gIMble simulate and two extreme barrier definitions: the complete absence of gene flow ($m_e^* = 0$, i.e. $\Delta_{B,0}$), or a very relaxed definition as a reduction in $m_e$ by half ($m_e^* < \hat{m}_e/2$, i.e. $\Delta_{B,0.5}$). Note that the latter is even more permissive than the definition of a weak barrier that we used in the *Heliconius* scan ($\Delta_{B,0.2}$). Here, the split time *T* was fixed to the background value while $N_e$ parameters were allowed to vary among windows. We assumed an average per-base and per-generation recombination rate of $r_{bp} = 1.89 \times 10^{-8}$ (given a male map length of 1083 cM and a genome size of 272.6 Mb [67]). ROC curves were constructed using 1,000 replicate simulations under the background IM model as true negatives (Table 2) and the model with a reduced local $m_e$ as true positives ($m_i = 0$ or $\hat{m}_e/2$).

Fig M in S1 Text shows the relationship between the recombination rate and the power to detect barriers. The impact of recombination is two-fold: on the one hand, with an increase in recombination rate we expect to see more topologies within a single window. On the other hand, an increase in recombination for the same block size increases the probability of observing four-gamete-test violations. Blocks containing mutation configurations that cannot be placed on a single topology are non-informative in this framework. Consistently leaving out a

particular subset of topologies might however bias the inference. Increasing the recombination rate therefore increases power but also leads to bias.

## Supporting information

**S1 Text. Supplementary methods, tables & figures.** The supporting information is organized into three sections: (1) Supplementary methods; `gIMble` commands used for the *Heliconius* analysis. (2) Supplementary tables A—C (3) Supplementary Figures A—M. (PDF)

## Acknowledgments

We thank Christelle Fraïsse and Stuart Baird for discussions and for critical comments on the manuscript and Richard Merrill for sharing the QTL results for female preference. We thank Alex Mackintosh, Noora Poikela, and Sam Ebdon for many helpful discussions and their patience when testing numerous pilot versions of `gIMble` and uncovering countless bugs. We are grateful to Mark Ravinet and the organisers of the SMBE satellite meeting on speciation genomics (Tjärnö 2019) for inviting us to present a pilot version of this method.

## Author Contributions

**Conceptualization:** Konrad Lohse.

**Data curation:** Dominik R. Laetsch, Gertjan Bisschop, Konrad Lohse.

**Formal analysis:** Dominik R. Laetsch, Gertjan Bisschop, Simon Aeschbacher, Derek Setter, Konrad Lohse.

**Funding acquisition:** Konrad Lohse.

**Investigation:** Dominik R. Laetsch, Gertjan Bisschop, Simon H. Martin, Simon Aeschbacher, Derek Setter, Konrad Lohse.

**Methodology:** Dominik R. Laetsch, Gertjan Bisschop, Simon H. Martin, Simon Aeschbacher, Derek Setter, Konrad Lohse.

**Project administration:** Konrad Lohse.

**Resources:** Simon H. Martin, Simon Aeschbacher, Konrad Lohse.

**Software:** Dominik R. Laetsch, Gertjan Bisschop.

**Supervision:** Konrad Lohse.

**Validation:** Dominik R. Laetsch, Gertjan Bisschop, Konrad Lohse.

**Visualization:** Dominik R. Laetsch, Gertjan Bisschop, Simon Aeschbacher, Derek Setter, Konrad Lohse.

**Writing – original draft:** Dominik R. Laetsch, Gertjan Bisschop, Simon H. Martin, Simon Aeschbacher, Derek Setter, Konrad Lohse.

**Writing – review & editing:** Dominik R. Laetsch, Gertjan Bisschop, Simon H. Martin, Simon Aeschbacher, Derek Setter, Konrad Lohse.

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
