## [Decision Letter · Decision Letter 0]

2 Aug 2023

Dear Dr Lohse,

Thank you very much for submitting your Research Article entitled 'Demographically explicit scans for barriers to gene flow using gIMble' to PLOS Genetics.

The manuscript was fully evaluated at the editorial level and by independent peer reviewers. The reviewers appreciated the attention to an important topic but identified some concerns that we ask you address in a revised manuscript. As noted below, the most significant concerns have to do with the accessibility of the paper to a non-technical audience. 

We therefore ask you to modify the manuscript according to the review recommendations. Your revisions should address the specific points made by each reviewer.

Yours sincerely,

Nicolas Bierne

Academic Editor

PLOS Genetics

Kelly Dyer

Section Editor

PLOS Genetics

Dear Dr Lohse,

I have received three thoughtful reviews of your manuscript, and all three reviewers agree that the gIMble model-based inference method is novel, timely, and worthy of publication. However, all three referees also urge you to make an effort to better describe the methodology, with the idea of improving clarity and achieving a broader impact. I would ask you to consider each reviewer's suggestions in a revised version.

Thank you for choosing PLoS Genetics to submit your original work.

Best regards,

Nicolas Bierne

Reviewer's Responses to Questions

**Comments to the Authors:**

Reviewer #1: The authors presented here a new method to detect genetic barriers between recently diverged species. The method builds upon their previous work, using coalescent theory and the site-frequency spectrum as the main statistic. The method uses a two-step approach: first population sizes, divergence time and gene flow between the two species are estimated genome wide. In a second time, all parameters but divergence time are estimated locally, with regions with a local migration rate of 0 (or below a certain threshold) interpreted as harboring genetic barriers. The method is illustrated in the case of Heliconius, with the authors recovering regions already known for harboring genetic barriers to gene flow.

The authors presented here a new approach that is nicely illustrated using Heliconius. The manuscript is in general clear and well written (but see point below).

Comments:

The authors followed the guidelines of PlosGen to have the methods at the end of the manuscript, but in my opinion, this decreases the readability of the manuscript. I would consider having a small model section first to explain the model being used here (i.e. how blocks and windows are defined and what the bSFS is.). In general, I found the details on how the different elements were defined a bit sparse.

- It was unclear to me how blocks were defined in the first place, in particular, whether the 64 bp have to be contiguous or not. I only found the answer that they do not have to be on line 698, because of the existence of the block_span option in addition to the block_length one. In addition, it is also not specified whether there is a minimum level of polymorphism expected for a given block. Within a block, can 30 bp be skipped because of coverage issue in one individual, or is it only indels and multiallelic sites that are skipped. Based on the definition of the bSFS (l 617 to 622), having a block without any polymorphism seems impossible, but it is unclear to me how such case is handled.

- Windows are defined over 500 blocks – but not over an uninterrupted stretch of DNA. Are there any conditions on the “missing” parts or can it be anything (including genes, small inversions, hot spot of recombination…). As a follow-up, can we therefore have genes within a window, or can they only be found outside windows.

- It would be nice to give a quick explanation (as in line 617 to 622) of what is the bSFS when it is introduced.

- I would consider adding a scheme with how blocks, windows and bSFS are related as a figure or a supplement. It would help the reader to understand the underlying logic of the approach.

I understand the logic of doing a 2-step estimation. However, I wonder whether the authors could test the reliability of this approach through subsampling or jackknifing of the different chromosomes. How variable is the estimate of T, since further down analysis relies on it being a global parameter.

It is unclear to me how sensitive the method is to the mutation rate. I am wondering how an increase/reduction in local mu may be reflected on the local estimates of m or N.

I wonder how much the grid has an impact on the result. I understand the computational burden and the impossibility of using a larger grid, but I wonder how robust the estimations obtained are to a subsampling of the grid (since the grid is precomputed, it should be doable).

Finally, I wonder whether the author could discuss how dependent the approach is to having gene flow in one direction only. In the example provided, backcrosses in one direction are impossible, but what happens if one direction is negligible compared to the other. Lets call m12 migration in the “main” direction and m21 migration in the opposite direction. Here m21=0. But what happens if m21= 0.01 m12. Is the method still applicable? What about m21=1/1000 m12 or 1/10?

Minor comments:

L178: “Crucially, we assume that blocks are only indirectly affected by selection at nearby linked sites and otherwise evolve neutrally and under a constant mutation rate. While these assumptions allow us to treat blocks within the same window as statistically exchangeable, they necessitate a careful and consistent filtering strategy for variant and invariant sites.” I am a bit confused here. LD is still going to decay as a function of the distance from the selected sites, especially if genes can only be found outside windows (and not within but skipped).

L208 and 209: are windows defined over 50,000 or 500 blocks?

L208: I would precise here that by offset, the authors mean that the window is moved by 10,000 pair blocks (and therefore windows overlaps) and not by 60,000 pair blocks (and windows don’t overlap).

L242-243: Could this variation not be explained by changes in recombination or mutation rate?

L247: is this not always true, even if there were no local variation in any of the parameters of the model. The local fitting simply captures part of the noise in that case. If you were to do the fitting of the parametric bootstrap data (if doable), using the same approach allowing for local variation, how much would the fit of the model be improved?

L 255: it is unclear to me how conservative this “normalization” is. Assuming Ne was truly constant, how much of the variation due to changes in local me would be captured by this approach?

Figure 4 : what is an effective length?

Figure S1: what are na and nb?

Reviewer #2: attached

Reviewer #3: The gIMble approach has already gained considerable popularity in the population genomics community over the last few years, and its publication is eagerly awaited (as it has already been used/approved by some). So I'm not going to delay its publication, or create a false suspense in my review: the article must be published, I support it 100%.

**methodological dissemination**

Having said that, I have my doubts about the current form of the article, particularly in terms of making it more accessible to non-methodologists. In my opinion, the article lacks a textbook-style figure. For example, DaDi is so popular because of the visual simplicity of the original article that, in 2023, everyone associates SFS with DaDi, simply because the link between demographic history, observed SFS and computational inference is accessible to the greatest number of people (even if the heart of the method is a black box for many readers).

Of course, Figure 1 is an attempt at a synthesis, but I don't understand it: what's a grid? What's the difference between green, orange, blue and purple? Why are there dotted lines between the purple grids? What does the gIMble store actually contain? What are orange, blue and violet spectra? Etc...

This figure is too 'bioinformatician' and didn't interest me; I think it would be better placed in a Readme. On the other hand, given that the authors are using a lexical corpus that is not very widespread in the literature, I think it is important to explain better what a tally is, what a grid is, the link between tally and bSFS, and how a barrier bSFS is different from a non-barrier bSFS.

Of course, this is mentioned in the text, but instead of the current figure 1, I can imagine a figure that would be used in presentations by people using gIMble, a more schematic figure based on an alignment of sequences obtained in species A and B. Part of this alignment is linked to a barrier (central part, for example), and the rest is not linked to a barrier.

Then to show the reader graphically and sequentially how gIMble handles such an alignment. This will give the reader a better understanding of what a block, a grid, a tally and a bSFS are, and how these elements can be used to distinguish a barrier region from a non-barrier region.

This is just a suggested figure, but I think it will increase the impact of the article more than the current figure 1, which doesn't really help you understand how gIMble works (it's more about the workflow architecture with a black box at its centre).

**Practical application to empirical data**

The application with the _Heliconius_ data is well written overall, but some additional information could be added to help the future user make choices.

1. In particular, how do you choose the length of the blocks in nucleotides? Why/how did the authors choose 64? Is this an important parameter in the analysis, and how would the results differ with different block sizes?

2. how flexible is gIMble with regard to the scenarios to be compared, i.e. which scenarios will users actually be able to compare? For example, is bidirectional introgression really impossible to study (with asymmetric barriers)?

3. fig. 2, grey pannels: why doesn't it look like I have 16 bins between 0 and 1e-6?

4. concerning the choice of thresholds for the migration rate, 0.2 times the average rate corresponds roughly to 0.08 migrants per generation in the _Heliconius_ example. But in some cases, 0.2 times m_e may correspond to a greater number of migrants per generation, whose effects on coalescence are the same as 1 times m_e. In order to define the m_e threshold, shouldn't readers be given a threshold expressed directly in terms of the number of migrants per generation?

5. lines 275-280: for me, the fact that barriers are not just Fst outliers is a very important message to get across to the community. I'm dreaming of a synthesis article where this would be highlighted for different types of living organisms (plants, animals, fungi).

6. In the "Robustness to model choice" paragraph, without wanting to come across as a tiresome reviewer with all the potential biases, it is difficult to ignore the secondary contacts. I think that future users need to be reassured here about the impact that such a model would have on gIMble inferences. This can also be done by providing information on how to process such a dataset with gIMble.

7. without checking whether this is the case or not, if it isn't, I think it would be useful if the VCFs (or fasta) used to analyse the _Heliconius_ data could be downloaded directly rather than having to redo all the bioinformatics. This dataset is perfect for users to get to grips with the tool, or to compare gIMble results with other t.q DILS methods, for example.

8. Finally, a more personal question: I'm currently interested in comparing speciation dynamics between contrasting groups of organisms (e.g. plants _versus_ animals; internal _versus_ external fertilization; parasites _versus_ non-parasites).

How optimistic would the authors of gIMble be about applying this tool to such analyses involving several dozen pairs of species?

Once again, I have only superficial comments to make, as I'm already convinced of the relevance and quality of the tool. I simply think that this article needs to be made more like a textbook than a semi-readme, which is currently too technical to understand the basics of gIMble, and also too superficial to really do a step-by-step analysis (which is not the aim of a scientific article).

**Have all data underlying the figures and results presented in the manuscript been provided?**

Reviewer #1: Yes

Reviewer #2: Yes

Reviewer #3: Yes

PLOS authors have the option to publish the peer review history of their article (what does this mean?). If published, this will include your full peer review and any attached files.

Reviewer #1: No

Reviewer #2: No

Reviewer #3: No

---

## [Editor Report · Decision Letter 1]

25 Sep 2023

Dear Dr Lohse,

We are pleased to inform you that your manuscript entitled "Demographically explicit scans for barriers to gene flow using gIMble" has been editorially accepted for publication in PLOS Genetics. Congratulations!

Yours sincerely,

Nicolas Bierne

Academic Editor

PLOS Genetics

Kelly Dyer

Section Editor

PLOS Genetics

Comments from the reviewers (if applicable):

Dear Dr Lohse,

After a careful reading of your revised manuscript and the associated response to the reviewers, I have decided that you have done a good job of revision and have accurately addressed each of the reviewers' concerns. There is no need for further review and I'm happy to recommend acceptance of your manuscript for publication in PLOS Genetics.

Thank you for choosing PLOS Genetics to publish your interesting method and work. I expect it to be an important contribution to the field.

Best regards,

Nicolas Bierne

**Data Deposition**

http://datadryad.org/submit?journalID=pgenetics&manu=PGENETICS-D-23-00626R1

**Press Queries**

---

## [Editor Report · Acceptance letter]

4 Oct 2023

PGENETICS-D-23-00626R1 

Demographically explicit scans for barriers to gene flow using gIMble 

Dear Dr Lohse, 

We are pleased to inform you that your manuscript entitled "Demographically explicit scans for barriers to gene flow using gIMble" has been formally accepted for publication in PLOS Genetics! Your manuscript is now with our production department and you will be notified of the publication date in due course.

With kind regards,

Anita Estes

PLOS Genetics

On behalf of:
